# Rapid learning with phase-change memory-based in-memory computing through learning-to-learn

Thomas Ortner [1,4], Horst Petschenig [2,4], Athanasios Vasilopoulos [1], Roland Renner[2], Špela Brglez [2], Thomas Limbacher[2], Enrique Piñero[3], Alejandro Linares-Barranco [3], Angeliki Pantazi [1] & Robert Legenstein [2]

There is a growing demand for low-power, autonomously learning artificial intelligence (AI) systems that can be applied at the edge and rapidly adapt to the specific situation at deployment site. However, current AI models struggle in such scenarios, often requiring extensive fine-tuning, computational resources, and data. In contrast, humans can effortlessly adjust to new tasks by transferring knowledge from related ones. The concept of learning-to-learn (L2L) mimics this process and enables AI models to rapidly adapt with only little computational effort and data. In-memory computing neuromorphic hardware (NMHW) is inspired by the brain's operating principles and mimics its physical co-location of memory and compute. In this work, we pair L2L with in-memory computing NMHW based on phase-change memory devices to build efficient AI models that can rapidly adapt to new tasks. We demonstrate the versatility of our approach in two scenarios: a convolutional neural network performing image classification and a biologically-inspired spiking neural network generating motor commands for a real robotic arm. Both models rapidly learn with few parameter updates. Deployed on the NMHW, they perform on-par with their software equivalents. Moreover, meta-training of these models can be performed in software with high-precision, alleviating the need for accurate hardware models.

Contemporary artificial intelligence (AI) models often rely on deep learning[1,2], resulting in intense computational requirements that become increasingly difficult to fulfill with current technology. This stands in contrast to a growing demand for low-power, autonomously learning AI systems that can be deployed at the edge, without access to large compute clusters. The number of applications for such systems is rapidly increasing and includes mobile devices, autonomous mobile robots and vehicles, smart sensors, and even the Internet of Things.

Due to its energy efficiency, neuromorphic hardware (NMHW) is a promising solution for these scenarios[3-7]. In particular, there were several recent breakthroughs in analog in-memory computing neuromorphic hardware systems[8-10]. They utilize analog memristive devices[11,12] arranged in a crossbar configuration, enabling the execution of matrix-vector multiplication (MVM)−the central operation in deep learning−in constant time, showcasing remarkable performance and efficiency. However, the limited precision of NMHW, attributed to device and circuit non-idealities, necessitates the adoption of hardware-aware training routines[13], chip-in-the-loop fine-tuning approaches[8,14,15], or the integration of accurate hardware models during training. Moreover, edge applications often demand online

[1]IBM Research Europe - Zurich, Rüschlikon, Switzerland. [2]Institute of Machine Learning and Neural Computation, Graz University of Technology, Graz, Austria. [3]Robotics and Technology of Computers, SCORE Laboratory EPS-ETSII, Universidad de Sevilla, Seville, Spain. [4]These authors contributed equally: Thomas Ortner, Horst Petschenig. ✉e-mail: robert.legenstein@tugraz.at

adaptation, yet neuromorphic hardware systems are typically tailored towards inference applications where no or only very little adaptation is needed, as these adaptations can compromise the high energy efficiency.

To equip neural networks with rapid learning capabilities, requiring only few adaptation steps and that are also robust to hardware non-idealities, we propose in this article the application of learning-to-learn (L2L; also known as meta-learning)[16–20] to neuromorphic hardware based on phase-change memory (PCM) devices[21]. In L2L, neural networks are first optimized to become good learners for a family of related tasks in a meta-training phase and in a subsequent adaptation phase tuned for a particular task, leveraging prior acquired knowledge. In contrast to standard learning approaches, where the neural network becomes specialized for one particular task, L2L through its two phases can enable rapid adaptation to a concrete task of the application that is a member of a larger family of tasks. The initial meta-training phase can be done off-chip with arbitrarily complex learning algorithms and large datasets, while in the concrete application, the meta-trained system can be adapted by updating only few parameters. Our approach is perfectly suited for PCM-based neuromorphic hardware, because (1) the network architecture can be trained in a meta-training phase in software, transferred once to the hardware, and then only a tiny fraction of the network needs to be updated during the adaptation phase, (2) the learning rule for the parameter updates during the adaptation phase becomes simple and could potentially be implemented directly on the neuromorphic hardware in future generations, and (3) the PCM devices are non-volatile, once adjusted to specific classes, the system can perform inference over an extended period of time.

Learning-to-learn has been widely studied in the field of machine learning (ML)[22–27]. However, although L2L has been proposed as an efficient way to enable rapid learning in neuromorphic systems[5], only few studies in this direction have been carried out so far[28,29]. Bohnstingl et al.[28] evaluated an L2L approach using a single-layer spiking neural network on basic Markov decision processes and a bandit task. Wu et al.[29] proposed a hybrid learning approach where parameters of local plasticity rules are meta-learned and combined with global plasticity. They implemented their model on the Tianjic neuromorphic platform[30]. Studies, where L2L was applied to simulated memristor models, are described in refs. 31,32. Zhang et al.[31] demonstrate an L2L algorithm for artificial neural networks and demonstrate it on the Omniglot and the MiniImageNet datasets. However, the authors utilize a software tool to simulate the PCM devices and also update the entire network architecture during meta-training and task adaptation. In the Supplementary information, we added a detailed table, comparing related works leveraging learning-to-learn and neuromorphic hardware. To the best of our knowledge, no application of L2L to physical memristor-based in-memory neuromorphic hardware has been reported so far.

To demonstrate the versatility of our approach we evaluated two L2L methods in two types of neural network models using the PCM-based in-memory computing neuromorphic hardware introduced in ref. 21. First, we applied model-agnostic meta-learning (MAML)[22], an L2L algorithm that optimizes initial weights of a neural network to enable few-shot adaptation of the network with a small number of gradient updates, to a convolutional neural network (CNN) for few-shot image classification. Since this algorithm necessitates extensive meta-training, we perform the meta-training phase in simulations and transfer the resulting weight values to the memristive crossbar. The adaptation phase is then performed directly on the neuromorphic hardware. Evaluations on the Omniglot dataset[33] showed that this approach leads to excellent classification performance, on par with pure software solutions, despite the fact that synapses are realized with low-precision PCM devices. Interestingly, our results also show that meta-training is quite robust with respect to the software model of

the hardware. In particular, we found that expensive and slow hardware-in-the-loop training is not necessary, and even a relatively crude software approximation of the hardware achieved good results, alleviating the need for accurate hardware models.

To complement the ML scenario, in the second application, we considered a recurrent spiking neural network (SNN). Recurrent SNNs are of particular interest since spike-based communication is highly energy efficient, and therefore a promising alternative to the energy-demanding contemporary AI solutions[34,35]. One fundamental problem in the application of recurrent neural networks in edge applications is that standard gradient-based learning algorithms, such as error back-propagation through time (BPTT), are not well-suited as they cannot simultaneously process and learn from incoming data streams. To address these issues, the research community has developed online learning alternatives to BPTT[36,37], that equip neural networks with this capability. Therefore, in this work we pair L2L with an online learning algorithm, e-prop[38], to build an energy-efficient system utilizing PCM-based neuromorphic hardware, that can rapidly adapt to new tasks online. In this algorithm, a teacher SNN, the Learning signal generator (LSG), generates learning signals that are used to update the weights of a second SNN called the trainee. In the meta-training phase, the initial weights of the trainee as well as the weights of the LSG are optimized. Then the initial weights of the trainee are ported to the neuromorphic hardware and a single update is performed to adapt the trainee to the current task. We used this setup to enable the SNN to learn to generate motor commands for a robotic arm to produce a target trajectory from a single exposure as proposed in refs. 36,38. In addition to simulations and experiments with the neuromorphic hardware, we also tested the model with a real robotic setup. Underpinning our findings of the first task, we found that meta-training can be performed in full-precision software, without the need for detailed hardware models, and that the single update on the neuromorphic hardware allows the robot to accurately track the target trajectory.

## Results
### Learning-to-learn and neuromorphic hardware
Learning-to-learn is a technique that aims to generalize the learning processes across multiple related tasks from a distribution of tasks, often termed the task family. It is based on the observation that in humans and animals, learning generally is not centered solely around acquiring knowledge or skills for a specific task, but rather on the development of strategies that enable learning new skills both more effectively and efficiently in the future[20,39,40]. Therefore, meta-learning aims to consolidate previously gained experience from different tasks to enable more rapid learning of new tasks requiring minimal new data.

In contrast to other approaches, such as standard supervised learning, meta-learning is carried out in two phases, the meta-training phase and the adaptation phase, see Fig. 1a. In the meta-training phase, which is performed in software, indicated with green color in the left part of Fig. 1a, all parameters of the neural network are trained through $m$ iterations of a meta-training procedure that includes both an outer training loop and an inner training loop (see below). After meta-training, the model is deployed onto, potentially many, neuromorphic hardware instances, where all weights of the network are mapped to the hardware, see yellow color in the right part of Fig. 1a. In the following adaptation phase, each hardware instance is adapted to an individual task from the task family. In this phase, the parameter update algorithm is very simple (it is given by the inner training loop only, as described below) and these updates are performed directly on the NMHW, to adapt a small fraction of the model for each individual task (indicated with the dashed rectangle in Fig. 1a).

The two levels of optimization used in the meta-training phase, the inner and the outer loop are illustrated in Fig. 1b and in more detail in Fig. 1c. They can be described as follows:

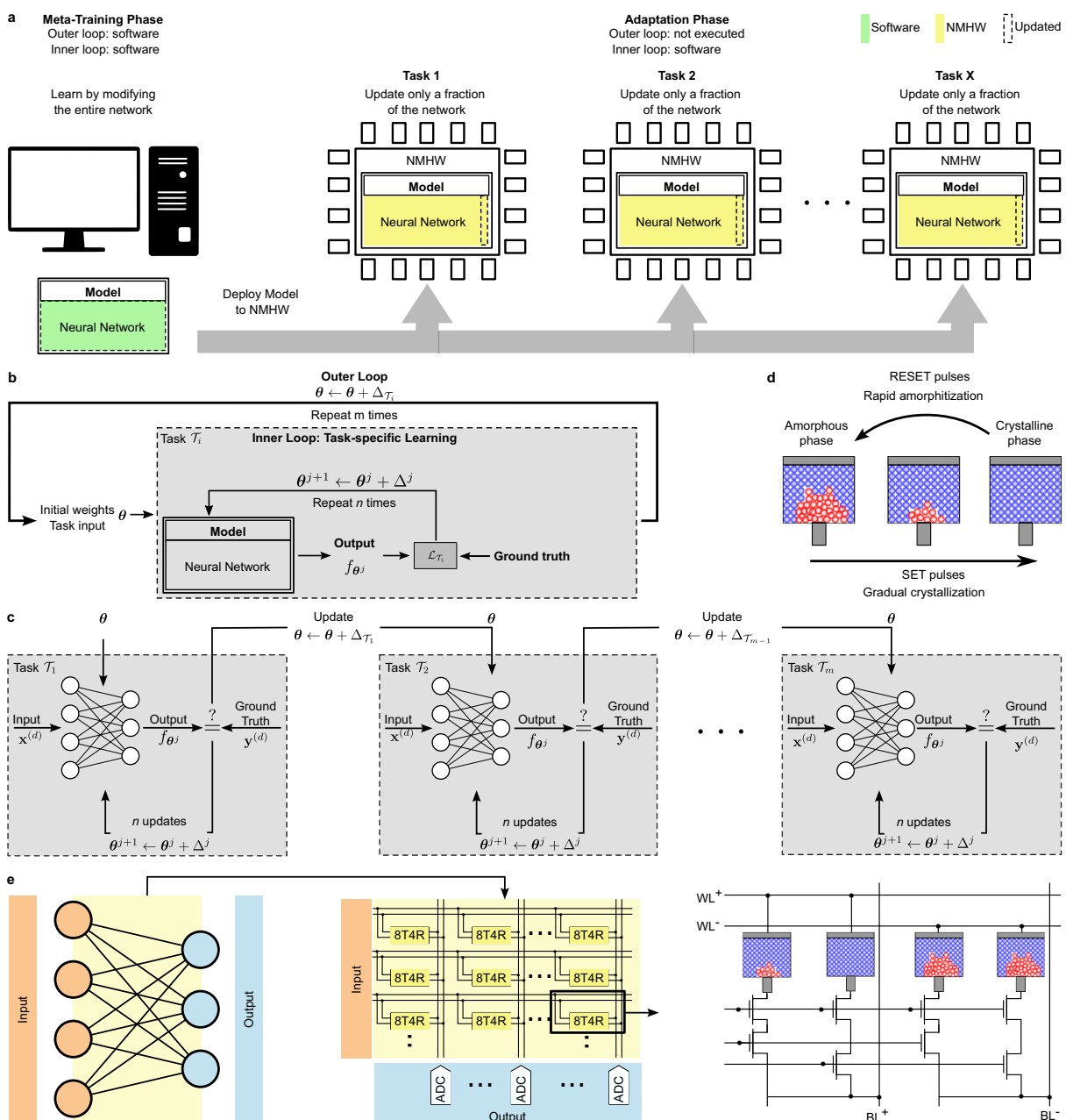

**Fig. 1 | Overview of learning-to-learn with neuromorphic hardware. a** General training strategy. Left: the meta-training phase is performed with all network weights kept in software (green), involves many iterations, and all weights are updated (dashed rectangle). Right: after meta-training, the model is deployed onto, potentially many, NMHW instances, where the weights are located on the hardware (yellow). Then, the adaptation phase is carried out, performing task-specific adaptation involving only a few iterations of a simple learning procedure. Importantly, this process only updates a fraction of the weights of the network architecture (dashed rectangle), and the updates are performed on the NMHW, without requiring a re-mapping of the entire network. **b** The general structure of meta-learning approaches used in this article. The inputs to the inner loop (gray box) are the initial parameters $\theta$ and the task inputs from a task $\mathcal{T}_i$. Based on these inputs, a fraction of the neural network is updated $n$ times. In every iteration of the outer loop, a new task $\mathcal{T}_i$ from the task family $\mathcal{F}(\mathcal{T})$ is selected, the inner loop is executed,

and the initial parameters $\theta$ are updated based on the errors of the inner loop. The goal is to find initial parameters $\theta$ such that a few inner loop updates lead to good results on any task from $\mathcal{F}(\mathcal{T})$. **c** Unrolled meta-learning procedure that highlights the differences between task-specific adaptation of weights in the inner loop and the meta-parameters in the outer loop. **d** Schematic depiction of a phase-change memory device and its inner workings. Information is stored in the phase configuration of the material and electrical pulses can be used to switch between the amorphous and the crystalline phase. **e** The employed NMHW comprises a crossbar array structure where at each intersection four PCM devices (4R) and eight control transistors (8T) are located. Two PCM devices represent positive weights (BL$^+$) and two represent negative weights (BL$^-$). The weights of a neural network are mapped onto the crossbar structure and the network inputs are provided to the positive devices using (WL$^+$) and to the negative devices using (WL$^-$).

Inner loop (task-specific learning): This part of the meta-learning procedure, illustrated in the gray box of Fig. 1b, is responsible for learning a specific task. While relying on previous experience from other related tasks, the model in the inner loop learns the current task with access to a limited set of training examples. The goal is to improve

task-specific performance via fast adaptation of the model parameters $\theta$ through $n$ updates.

Outer loop: in the outer loop, indicated by the outer black arrow in Fig. 1b, the model learns across multiple tasks with the goal of identifying common sub-structures and potential differences between

them. This is done by adapting the meta-parameters and implicit learning strategies based on the models output and the task performance observed during the inner loop. Thus, instead of optimizing the performance for an individual task, the goal of the outer loop is to improve the ability of the model to adapt to new tasks more effectively and efficiently.

Different meta-learning methods have been developed and they can broadly be classified into model-based methods, initialization-based methods, and parameter-generation-based methods.

Model-based methods include memory-augmented model architectures or external memory modules that are inherently well-suited for learning from a limited amount of data. Memory-augmented neural networks[25] use a differentiable external memory module to store and fetch information from a small number of previously seen examples enabling rapid adaptation to new tasks.

Initialization-based methods are centered around the idea that there exists a learnable initialization of the model parameters that allow fast adaptation to new, unseen tasks. Model-agnostic meta-learning[22] is a method aiming to find initial model parameters that can be efficiently updated with a small number of gradient steps on new tasks.

Parameter-generation-based methods train networks that generate and predict parameters for a trainee network, enabling adaptation to new, unseen tasks. This can be facilitated in two ways: Either the parameters are generated directly and fed into the trainee network[23] or indirectly by learning an optimizer that can then change the trainee network's parameters[24,27].

In this work, we apply L2L techniques to an in-memory computing neuromorphic hardware, utilizing PCM devices. Phase-change materials belong to a class of materials that allow to storage information in their phase configuration. When electrical pulses are applied to the cells, they can gradually transition from an amorphous phase to a crystalline phase, or rapidly back from the crystalline phase to the amorphous phase, illustrated in Fig. 1d. The neuromorphic platform we used in this work comprises two computational cores. Each core contains a crossbar array structure of size $256 \times 256$, where at each intersection 4 PCM devices (4R) and eight control transistors (8T) are located. The weights of the various neural networks used in this work are mapped onto this crossbar structure as illustrated in Fig. 1e. More details about the hardware can be found in the Section "Neuromorphic hardware" in "Methods".

In particular, we investigated two different L2L methods applied to two different tasks to demonstrate rapid learning in PCM-based NMHW. In the first approach, we applied the initialization-based approach MAML to meta-train the weights of a convolutional neural network that could then be easily adapted to new tasks from the same domain. In the second approach, we utilized a parameter-generation-based method, which enabled a biologically-inspired spiking neural network to generate motor commands that produce a target trajectory using only a single adaption step.

### Few-shot image classification with PCM-based neuromorphic hardware

We first investigated whether L2L could be utilized to enable few-shot image classification in a PCM-based NMHW. To this end, we utilized model-agnostic meta-learning and tested the system on the Omniglot dataset[33]. As an initialization-based L2L algorithm, the central idea behind MAML is to determine initial model weights such that they can be adapted to a new task using only a small number of weight updates. This approach is model-agnostic insofar as it can be used for any model that can be trained with gradient-based algorithms.

As described in Section "Learning-to-learn and neuromorphic hardware", the training is carried out in two nested loops: the outer loop and the inner loop, see Fig. 2a. In the inner loop (gray box in Fig. 2a), the initial model parameters $\boldsymbol{\theta}^0$, obtained from the outer loop,

are updated for a specific task $\mathcal{T}_i$, sampled from the task family $\mathcal{F}(\mathcal{T})$. We denote the model parameters optimized by the outer loop with $\boldsymbol{\theta} \stackrel{\text{def}}{=} \boldsymbol{\theta}^0$. Further, we denote the parameters after the $j$-th inner loop update with $\boldsymbol{\theta}^j$ and the output of the model with parameters $\boldsymbol{\theta}^j$ as $f_{\boldsymbol{\theta}^j}$. For each new task $\mathcal{T}_i$, we have $N_{\text{data}}$ data points $\mathcal{D}_{\mathcal{T}_i} = \{\boldsymbol{x}^{(d)}, \boldsymbol{y}^{(d)}\}$ with network inputs $\boldsymbol{x}^{(d)}$ and corresponding targets $\boldsymbol{y}^{(d)}$. The adaptation process involves computing the task-specific updated parameters $\boldsymbol{\theta}^{j+1}$, using gradient descent on the loss $\mathcal{L}_{\mathcal{T}_i}(f_{\boldsymbol{\theta}^j})$, which compares the model output $f_{\boldsymbol{\theta}^j}$ to the targets $\boldsymbol{y}^{(d)}$. The inner loop update for a single step can thus be expressed as

$$\boldsymbol{\theta}^{j+1} = \boldsymbol{\theta}^j - \alpha \nabla_{\boldsymbol{\theta}^j} \mathcal{L}_{\mathcal{T}_i}(f_{\boldsymbol{\theta}^j}), \tag{1}$$

where $\alpha$ is the learning rate for the task-specific update. In our setting, this update is repeated $n = 4$ times in the inner loop.

**Algorithm 1**. Our model-agnostic meta-learning setup for few-shot classification. For the Omniglot task, the cross-entropy loss function is used.

> **Input:** $\mathcal{F}(\mathcal{T})$: Family of tasks
> **Input:** $\alpha, \beta$: learning rates
> **Input:** $n$: number of inner loop gradient steps
> Randomly initialize $\boldsymbol{\theta}$;
> **while** *Meta-Training* **do**
> Sample $N_{\text{tasks}}$ tasks $\mathcal{T}_i \sim \mathcal{F}(\mathcal{T})$;
> **foreach** $\mathcal{T}_i$ **do**
> Sample $N_{\text{data}}$ data points $\mathcal{D}_{\mathcal{T}_i} = \{\boldsymbol{x}^{(d)}, \boldsymbol{y}^{(d)}\}$ from $\mathcal{T}_i$;
> **for** $j$ from $0$ to $n - 1$ **do**
> Evaluate $\nabla_{\boldsymbol{\theta}^j} \mathcal{L}_{\mathcal{T}_i}(f_{\boldsymbol{\theta}^j})$ using $\mathcal{D}_{\mathcal{T}_i}$ and the cross-entropy loss $\mathcal{L}_{\mathcal{T}_i}$;
> Compute adapted parameters:
> $\boldsymbol{\theta}^{j+1} \leftarrow \boldsymbol{\theta}^j - \alpha \nabla_{\boldsymbol{\theta}^j} \mathcal{L}_{\mathcal{T}_i}(f_{\boldsymbol{\theta}^j})$;
> **end**
> Evaluate data points $\mathcal{D}'_{\mathcal{T}_i} = \{\boldsymbol{x}^{(d)}, \boldsymbol{y}^{(d)}\}$ from $\mathcal{T}_i$ using $\boldsymbol{\theta}^n$ for the outer loop update.
> **end**
> Update $\boldsymbol{\theta} \leftarrow \boldsymbol{\theta} - \beta \nabla_{\boldsymbol{\theta}} \sum_i \mathcal{L}_{\mathcal{T}_i}(f_{\boldsymbol{\theta}^n})$ using each $\mathcal{D}'_{\mathcal{T}_i}$;
> **end**

In the outer loop (outer black arrow in Fig. 2a), the initial parameters $\boldsymbol{\theta}$ of the model are then optimized such that learning of new, unseen data $\mathcal{D}_{\mathcal{T}_i}$ from the same tasks $\mathcal{T}_i$ in the inner loop is more efficient. Therefore, the meta-training objective can be formally expressed as

$$\boldsymbol{\theta} = \arg\min_{\boldsymbol{\theta}} \sum_{\mathcal{T}_i \sim \mathcal{F}(\mathcal{T})} \mathcal{L}_{\mathcal{T}_i}(f_{\boldsymbol{\theta}^n}), \tag{2}$$

where $\boldsymbol{\theta}^n$ refers to the parameters after the last inner loop update. Note that $\boldsymbol{\theta}^n$ depends implicitly on $\boldsymbol{\theta}$. This optimization problem is solved using the ADAM optimizer[41] across unseen tasks sampled from a task distribution $\mathcal{F}(\mathcal{T})$. See "Algorithm 1" for a detailed description of the interplay between meta-training and evaluation. Intuitively, this combination of outer and inner loop updates creates a path traversing through the parameter space, which is visually depicted in Fig. 2b. The initial parameters $\boldsymbol{\theta}$ provide a good starting point for all tasks in the task family $\mathcal{F}(\mathcal{T})$, which is then adjusted to the specific task $\mathcal{T}_i$ with only four parameter updates leading to $\boldsymbol{\theta}^1$, followed by $\boldsymbol{\theta}^2, \boldsymbol{\theta}^3$, and so on until $\boldsymbol{\theta}^n$. Further details on the learning algorithm can be found in the Section "Few-shot image classification" in "Methods".

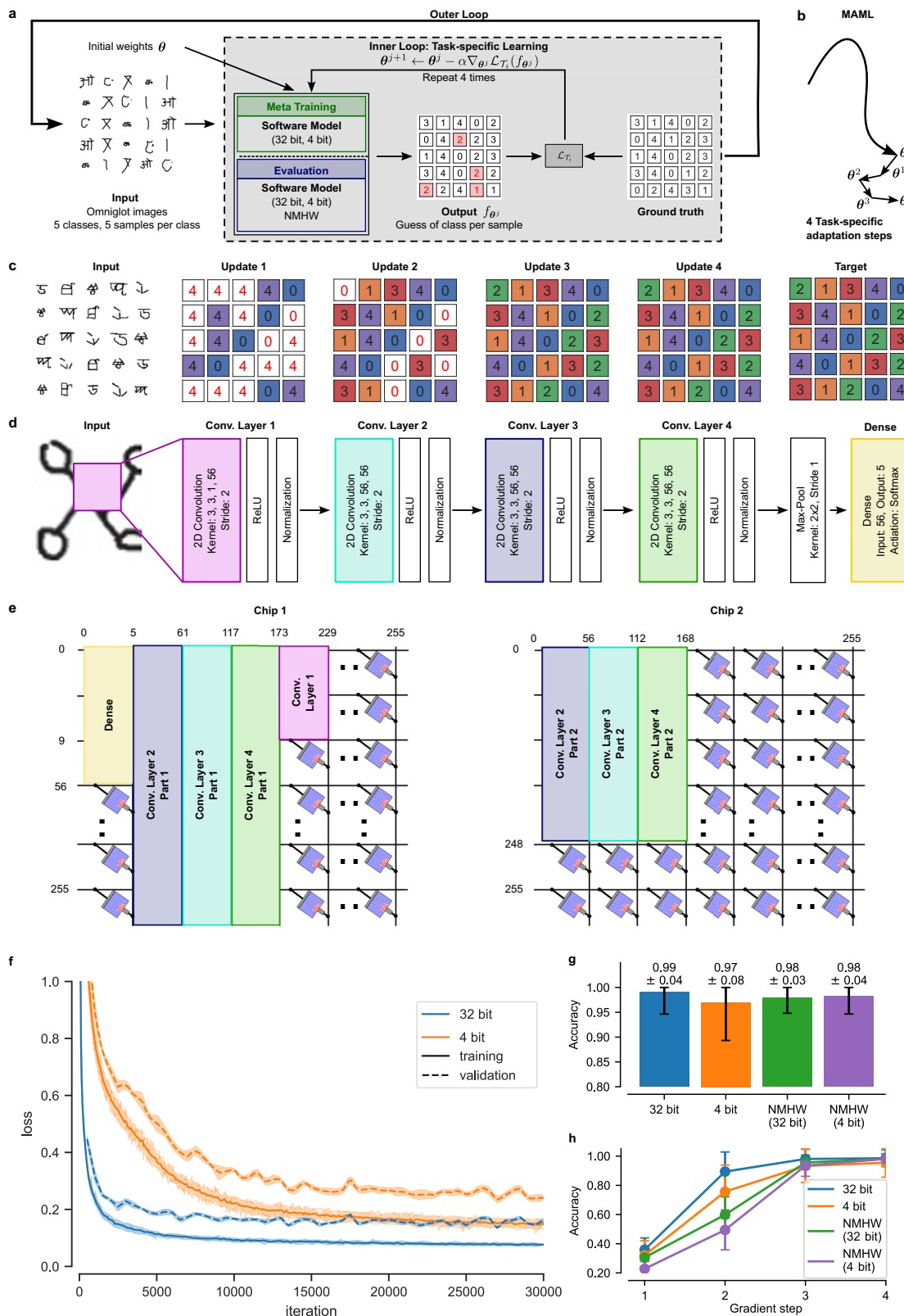

The Omniglot dataset[33] employed in this task is one of the most commonly used benchmark datasets for few-shot image classification that was specifically designed for understanding the few-shot learning capabilities of humans. The dataset contains 1623 grayscale images of handwritten characters originating from 50 different alphabets, with only 20 examples per character. We closely followed the experimentation protocol from[22,42] that describes an $N$-way classification

problem with $K$ shots. Here, $K$ examples of $N$ different classes are provided to the model during a training step with the goal to classify new, unseen examples of the $N$ different classes.

In our setup, we performed 5-way 5-shot classification with $n = 4$ gradient update steps in the inner loop. To illustrate the progress of the inner loop, we show in Fig. 2c the classification of the NMHW-based model after the individual updates. In particular, five new character

**Fig. 2 | Few-shot image classification on Omniglot with MAML. a** Illustration of the inner and outer loops in the MAML setup. In the inner loop, a software model was used for meta-training. The evaluation was performed both in software and in NMHW. For the inner loop training, we performed four gradient updates. **b** Schematic depiction of the movement in parameter space during MAML. The initial parameters $\boldsymbol{\theta}$ are optimized in the outer loop (bold trajectory) and the inner loop performs four task-specific adaptation steps (small arrows) such that the model achieves high classification accuracy. **c** Illustration of the input data from the Omniglot dataset for the 5-way 5-shot classification task on the left and the corresponding ground-truth targets on the right. A typical evolution of the classification performance of the model in the inner loop with 4 updates is illustrated in the middle. **d** Architecture of the four-layer convolutional neural network with a dense layer on top that is employed to solve the classification task. Only the dense layer, marked in yellow, is updated during the inner loop training, while the rest of the architecture remains fixed. **e** Schematic depiction of the mapping of the neural network to the NMHW. The convolutional layers are split into two parts and spread across the two crossbar arrays of the NMWH. **f** Evolution of the loss during outer loop training of a 4 bit (orange) and a 32 bit (blue) model in software. **g** Classification accuracy of the various models on 100 new unseen tasks. **h** Classification accuracy of the various models during inner loop training. Results with the label "NMHW" have been collected employing the NMHW described in Section "Neuromorphic hardware" in "Methods", with the mapping illustrated in (**e**).

classes were presented to the network, with five examples for each class in a random order, see Fig. 2c (left). The classification outputs after each of the four gradient updates in the inner loop are illustrated in Fig. 2c (middle): incorrect classifications are marked with red font on white squares and correct classifications with black font on colored squares. One can see that with each gradient update, the classification becomes more accurate.

We leveraged a CNN with four convolutional layers and a dense layer at the output, as depicted in Fig. 2d (see Section "Few-shot image classification" in "Methods" for a detailed network description). We mapped the individual kernels of the convolutional layers and the dense layer onto the crossbar arrays of the NMHW as illustrated in Fig. 2e. The convolutional kernels of the CNN were flattened, then split into two parts and distributed across two chips, see Section "Deploying models on the neuromorphic hardware" in "Methods" for further details. Although we mapped the entire CNN onto the NMHW, only the weights of the last dense layer, marked in yellow, in Fig. 2d were updated in the inner loop. These weights account for less than 1% of the entire CNN and thus dramatically simplify the inner loop training, as this approach avoids the need for backpropagation of gradients through the (neuromorphic) network. In particular, the update that has to be performed on the NMHW reduces to a simple delta rule[43]. More precisely, let $f_{\boldsymbol{\theta}^j, l}$ be the $l$-th output of the CNN, $y_l^{(d)}$ be the corresponding target for the $d$-th training example (where the class is indicated in standard one-hot encoding), and $h_k$ the $k$-th output of the max-pooling layer. Then the change of the corresponding weight $\theta_{lk}$ from the pooling layer to the output layer is given by

$$\Delta\theta_{lk} = \alpha\left(y_l^{(d)} - f_{\boldsymbol{\theta}^j, l}\right)h_k. \tag{3}$$

Meta-training typically necessitates a large number of training iterations. In our case, we used 30,000 outer loop iterations, which makes it infeasible to directly use the hardware during this phase. Instead, we carried out the meta-training phase entirely in software and did not consider hardware-aware training or accurate hardware models. To demonstrate that these techniques are indeed not needed with our approach, we also performed outer loop training with a limited weight-precision of 4 bit and later compared the classification accuracies. These 4 bit weights aim to simulate the neuromorphic hardware used in this work, as an effective 4 bit equivalent precision has been demonstrated in previous works[44,45]. In particular, we considered two cases: first, we trained a software model with 32 bit floating-point weights ("32 bit" setting). This model does not take into account the limited precision of PCM devices in the hardware. Second, we trained a network that employed 4 bit quantized weights with stochastic rounding ("4 bit" setting). Importantly, during outer loop training, no additional hardware-aware training technique was employed. Figure 2f shows the training and validation loss of the model during meta-training in software. The 32 bit version and the 4 bit version converged rather smoothly, and as expected the final training loss of the 4 bit model with $0.241 \pm 0.008$ was higher than the loss of the 32 bit model with $0.163 \pm 0.008$.

After meta-training we tested the few-shot learning capabilities of the models on 100 new unseen tasks (each task with 5 classes, i.e., 500 novel classes in total). In software, the achieved classification accuracies for the 32 bit and the 4 bit model were not significantly different, see left two bars in Fig. 2g. In addition to the evaluation of the software models, we also evaluated these models after porting them onto the NMHW, as described above. In the inner loop, inference was performed leveraging the crossbar arrays for efficient MVMs, updates for the dense layer were computed, and the corresponding weights were re-programmed onto the NMHW. For a detailed investigation of the weight distributions of the NMHW model, see Supplementary Fig. 1. Interestingly, the classification accuracies of the 32 bit and the 4 bit models ported onto the hardware were on-par with the software versions (Fig. 2g, right two bars). We can furthermore observe that the accuracy of the model that was trained on full-precision floating point weights in software was on-par with that of the model trained with 4 bit weights. Both results combined indicate that for this task, meta-training of a hardware-accurate model is not necessary. This is quite advantageous as one does not have to develop an accurate software model of the NMHW for meta-training.

As described above, we performed four consecutive updates of the network (each one containing a batch of 25 examples, 5 examples for each of the 5 classes). An analysis of the network performance after each individual gradient step is shown in Fig. h. We observe that the classification performance of the NMHW models during the first two gradient steps lacks behind the software models, but after the third and fourth gradient steps, the performance is on par. One example few-shot learning trial with NMHW (32 bit) is also shown in Fig. 2c. Moreover, in every gradient step only 1120 PCM devices, the dense layer, of a total of 342,720 PCM devices are updated, see Fig. 2e. Updating parameters stands out as one of the most energy-intensive operations and it could compromise the energy efficiency of the system—see Section "Neuromorphic hardware" in "Methods". Therefore, the rapid learning with only a small number of gradient steps alongside with the consideration that only a few PCM devices are updated in each step, proves particularly beneficial for NMHW.

We investigated whether a similar rapid learning performance can be obtained without the meta-training phase. Therefore, we used backpropagation (BP) in the adaptation phase and trained the same network from scratch on the 25 train images, and evaluated its generalization performance on the test images. Even though all models learn to classify the train images very quickly, i.e., with only very few gradient steps (Supplementary Fig. 2a), the BP-trained network exhibits significant overfitting leading to poor generalization performance to the test images (Supplementary Fig. 2b).

We also tested our approach on the more demanding CIFAR100-FS dataset[46], a few-shot learning dataset derived from CIFAR100. Similar to the case of the Omniglot dataset, Fig. 3a shows on the left the 5 input images from 5 classes, with their classes shown explicitly on the right. The middle part illustrates one example for the inner loop training with 5 update steps. For this more complex dataset, a bigger network architecture is required to achieve high accuracy[46]. In

particular, we employed a CNN with four convolutional layers with 256 hidden channels each and two dense layers with biases on top, one with 256 output units and 1 with 5. The architecture is schematically depicted in Fig. 3b. Importantly, similar as for the Omniglot dataset, we only trained the weights and the biases of the last dense layer, marked in yellow, during inner loop training. The weights are updated according to Eq. (3) and the biases are updated according to Eq. (11). Because this network architecture is too large to fit onto our prototype NMHW, we leveraged a hardware-accurate emulator of the PCM devices for our experiments[47], denoted with "EM-NMHW". This emulator was calibrated on one million PCM devices[48,49] and models all their major non-idealities, such as various noise effects. Therefore, it closely matches the behavior of the NMHW that we utilized and several works have already demonstrated equivalent accuracy between software, the emulator, and the real hardware[48,49]. Similarly, as for the Omniglot dataset, we evaluated two cases: first, a high-precision floating point model ("32 bit" setting) as well as a configuration with 4 bit quantized weights and stochastic rounding ("4 bit" setting). Importantly, during outer loop training, only pure software has been used and neither the emulator nor any other additional hardware-aware training technique was employed. Figure 3c shows the loss curves for the 32 bit and the 4 bit model during meta-training for 37,000 outer loop iterations. Compared to Fig. 2f, the loss curves exhibited more noise and slower convergence with final training losses of $0.353 \pm 0.008$ and $0.559 \pm 0.009$ for the 32 bit version and the 4 bit version, respectively. After meta-training we evaluated the models on 100 new unseen tasks (each task with 5 classes, i.e., 500 novel classes in total). Figure 3d shows the testing accuracy of the various models after the meta-training phase. As one can see, the emulated hardware model performed on-par with the full-precision software model. Again, 4 bit meta-training did not improve performance compared to meta-training with full precision. Figure 3e shows the accuracy evolution during inner loop training. We consistently observed accuracies close to 100% already after the fourth gradient step. Thus, we also evaluated the same "EM-NMHW (32 bit)" model with only 4 updates in the inner loop. We found that it performs on-par with the model that uses 5 updates, a green bar in Fig. 3, and a green curve in Fig. 3e.

In summary, these results showcase that L2L can be effectively applied to NMHW without the need for complex hardware models. Moreover, it enables rapid learning of new tasks using only 4 parameter updates.

## Rapid online learning of robot arm trajectories in biologically inspired neural networks

**Algorithm 2.** Natural e-prop for one-shot learning

> **Input:** $\mathcal{F}(\mathcal{T})$: family of tasks
> **Input:** $\alpha, \beta$: learning rates
> Randomly initialize $\boldsymbol{\theta}, \boldsymbol{\psi}$;
> **while** *Meta-Training* **do**
>     Sample $N_{\text{tasks}}$ tasks $\mathcal{T}_i \sim \mathcal{F}(\mathcal{T})$;
>     **foreach** $\mathcal{T}_i$ **do**
>         Compute trainee output $f_{\boldsymbol{\theta}}$;
>         Compute LSG output $\boldsymbol{L}^t$;
>         Compute eligibility trace $\boldsymbol{e}^t$;
>         Compute updated trainee parameters:
>         $\boldsymbol{\theta}^1 \leftarrow \boldsymbol{\theta} - \alpha \sum_t \boldsymbol{L}^t \odot \boldsymbol{e}^t$;
>         Evaluate task $\mathcal{T}_i$ using $\boldsymbol{\theta}^1$ for the meta update.
>     **end**
>     Update $\boldsymbol{\theta} \leftarrow \boldsymbol{\theta} - \beta \nabla_{\boldsymbol{\theta}} \sum_i \mathcal{L}_{\mathcal{T}_i}(\boldsymbol{\theta}^1, \boldsymbol{\psi})$;
>     Update $\boldsymbol{\psi} \leftarrow \boldsymbol{\psi} - \beta \nabla_{\boldsymbol{\psi}} \sum_i \mathcal{L}_{\mathcal{T}_i}(\boldsymbol{\theta}^1, \boldsymbol{\psi})$;
> **end**

Biology offers endless inspiration for the design of intelligent computing systems. In fact, the human brain exhibits unrivaled abilities in terms of learning from limited data and rapid adaptation to new tasks. It has been proposed that these capabilities arise from the utilization of prior accumulated knowledge through evolutionary processes and L2L has been used to model such capabilities[19]. In contrast to neurons in conventional artificial neural network models, biological neurons integrate synaptic inputs over time and communicate binary events, so-called spikes, to other neurons within recurrent networks. Mathematical models for such neurons, termed spiking neurons, have been developed[50] and spiking neural networks are networks composed of spiking neurons[50,51]. Therefore, we explored in a second experiment the applicability of L2L to recurrent SNNs with a biology-inspired meta-learning approach leveraging NMHW. In particular, we utilized the biologically plausible learning algorithm natural e-prop[36]. Natural e-prop was designed based on two observations about learning-related synaptic plasticity in the brain. First, molecular processes in synapses store information about local events such as pre- and post-synaptic activity that is relevant for future synaptic weight updates. These molecular traces are called eligibility traces[52]. Second, specialized brain areas produce learning signals that are communicated to synapses throughout the brain, for example in the form of dopamine release or neuronal firing[53]. In natural e-prop, these learning signals are generated in a dedicated SNN, the learning signal generator. The learning signals are communicated to the trainee, another SNN that adapts its synaptic weights based on these signals, and eligibility traces that are computed based on local signals at its synaptic connections. Fig. 4a shows an illustration of the L2L setup with natural e-prop and Fig. 4b shows a more detailed depiction of the LSG and trainee. Further details about the network architecture and the SNN dynamics can be found in the Section "One-shot learning via natural e-prop" in "Methods".

We denote the synaptic weights of the trainee and the LSG with $\boldsymbol{\theta}$ and $\boldsymbol{\psi}$, respectively. One inner loop trial consists of one simulation of the SNNs for $T$ time steps. In each time step $t$, the output of the LSG gives rise to one learning signal $L_l^t$ for each neuron $l$ in the trainee network. In the trainee, each synapse $lk$ from neuron $k$ to neuron $l$ updates its eligibility trace $e_{lk}^t$ (see Section "One-shot learning via natural e-prop" in "Methods"). The eligibility traces are combined with the learning signals to obtain the updated weights

$$\theta_{lk}^1 = \theta_{lk} - \alpha \sum_t L_l^t e_{lk}^t, \tag{4}$$

where $\theta_{lk}$ is the initial weight. To simplify the notation, let $\boldsymbol{\theta}$ denote the vector of all synaptic weights in the trainee, and $\boldsymbol{e}^t$ denote the vector of the corresponding eligibility traces at time $t$. We define the vector of learning signals $\boldsymbol{L}^t$, where $L_l^t$ is the learning signal corresponding to $\theta_l$. Then, we can write

$$\boldsymbol{\theta}^1 = \boldsymbol{\theta} - \alpha \sum_t \boldsymbol{L}^t \odot \boldsymbol{e}^t, \tag{5}$$

where $\odot$ denotes the component-wise product of the vectors. After the update, the trainee is run again (without any parameter changes) and its output is used to compute a loss.

In each outer loop training iteration, a task $\mathcal{T}_i \sim \mathcal{F}(\mathcal{T})$ is chosen, the inner loop is performed, and the parameters of the LSG as well as the initial weights of the trainee are updated based on the loss of the inner loop. The process of the inner and outer loop learning is illustrated in Fig. 4a, see also "Algorithm 2".

The meta-training objective is to find the optimal initial parameters $\boldsymbol{\theta}$ for the trainee and the parameters $\boldsymbol{\psi}$ of the LSG that minimize the average loss across several tasks, which can be

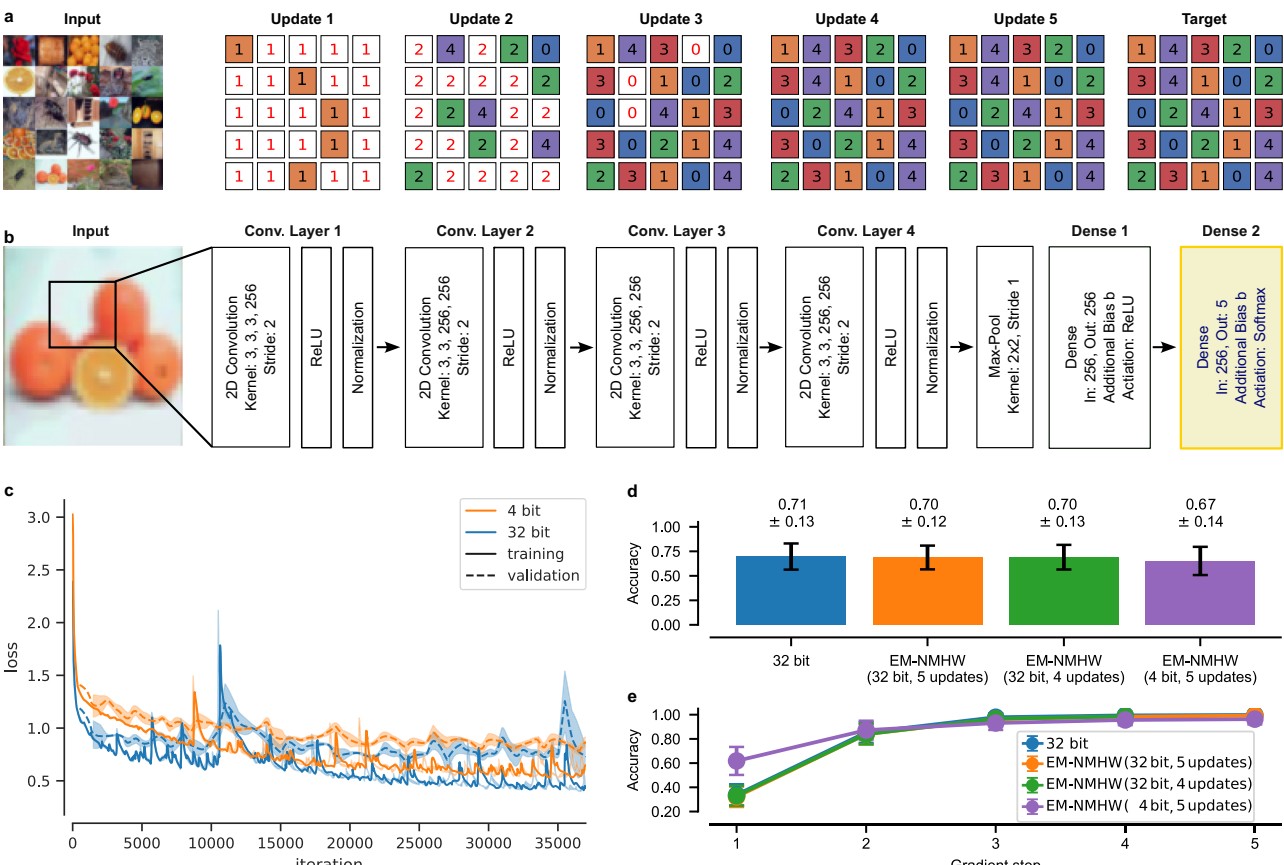

**Fig. 3 | Few-shot image classification on CIFAR100-FS with MAML. a** Illustration of the input data from the CIFAR100-FS dataset for the 5-way 5-shot classification task on the left and the corresponding ground-truth targets on the right. A typical evolution of the classification performance of the model in the inner loop with 5 updates is illustrated in the middle. **b** Architecture of the four-layer convolutional neural network with two dense layers on top. Only the last dense layer, marked in yellow, is updated during the inner loop training, while the rest of the architecture remains fixed. **c** Evolution of the loss during outer loop training of a 32 bit model (blue) and a 4 bit model (orange) in software. **d** Classification accuracy of the 32 bit software model and the emulated hardware model with 4 and 5 inner loop updates on 100 new unseen tasks. **e** Classification accuracy of the various models during inner loop training. Results with the label "NMHW" have been collected employing the NMHW described in Section "Neuromorphic hardware" in "Methods".

formally expressed as

$$\boldsymbol{\theta}, \boldsymbol{\psi} = \arg\min_{\boldsymbol{\theta},\boldsymbol{\psi}} \sum_{\mathcal{T}_i \sim \mathcal{F}(\mathcal{T})} \mathcal{L}_{\mathcal{T}_i}(\boldsymbol{\theta}^1) \tag{6}$$

$$= \arg\min_{\boldsymbol{\theta},\boldsymbol{\psi}} \sum_{\mathcal{T}_i \sim \mathcal{F}(\mathcal{T})} \mathcal{L}_{\mathcal{T}_i}\left(\boldsymbol{\theta} - \alpha \sum_t \boldsymbol{L}^t \odot \boldsymbol{e}^t\right). \tag{7}$$

Note that the learning signals $\boldsymbol{L}^t$ generated by the LSG depend on the parameters $\boldsymbol{\psi}$, whereas the eligibility trace $\boldsymbol{e}^t$ arising from the trainee depends on the parameters $\boldsymbol{\theta}$. We performed this optimization using the ADAM optimizer.

We tested this setup on the task to learn to generate motor commands for a robotic arm that produces a target trajectory from a single exposure as proposed in refs. 36,38. In addition to experiments with the neuromorphic hardware, we also tested the model in a real-world robotic scenario employing the ED-Scorbot robotic arm (see Fig. 4c and Section "The ED-Scorbot robotic arm" in "Methods").

In this task, the input to the trainee consisted of a clock-like input signal provided by five input neurons where the neurons were sequentially activated, each for 50 ms. This indicated the approximate temporal position within the 250 ms long target trajectory. The output of the trainee consisted of 2 neurons that encoded the motor commands, i.e., the angular velocities for two of the five joints of the robot, the base joint and the shoulder joint (see Fig. 4b). The input to the LSG

was the same clock signal plus the target 3D trajectory in Euclidean coordinates encoded by 53 input neurons. The LSG provided 250 learning signals through its output neurons (see Section "One-shot learning via natural e-prop" in "Methods" for details). The LSG and the trainee were first executed for 250 ms with these inputs, thus making the target trajectory available to the LSG. Then, the weights of the trainee were updated and the trainee was run for another 250 ms with the same clock-like input signal. The goal was that after this single update, the trainee controlled the robot such that it performed the target trajectory. Note that while the target trajectory was given in Euclidean coordinates, the output of the trainee was angular velocities for joints.

Again, meta-training was performed with a software model of the hardware, where we used full-precision floating point weights. In this phase, we also used a simulation model for the robot (see Section "The ED-Scorbot robotic arm" in "Methods" for details). In the inner loop update and during testing after meta-learning, only the input weights $\boldsymbol{\theta}^{\text{in}}$ and the recurrent weights $\boldsymbol{\theta}^{\text{rec}}$ of the trainee network were adapted. These weight matrices were mapped onto a single core of the NMHW after meta-training, see Fig. 4d. Non-plastic weights were kept in software.

Figure 5a shows the network output and the robot trajectory after meta-training but before the one-shot update. The angular velocity commands for both joints are depicted in the first two panels and the executed trajectory in the Euclidean space is depicted in the rightmost panel. The network output of the full-precision model (32 bit) is

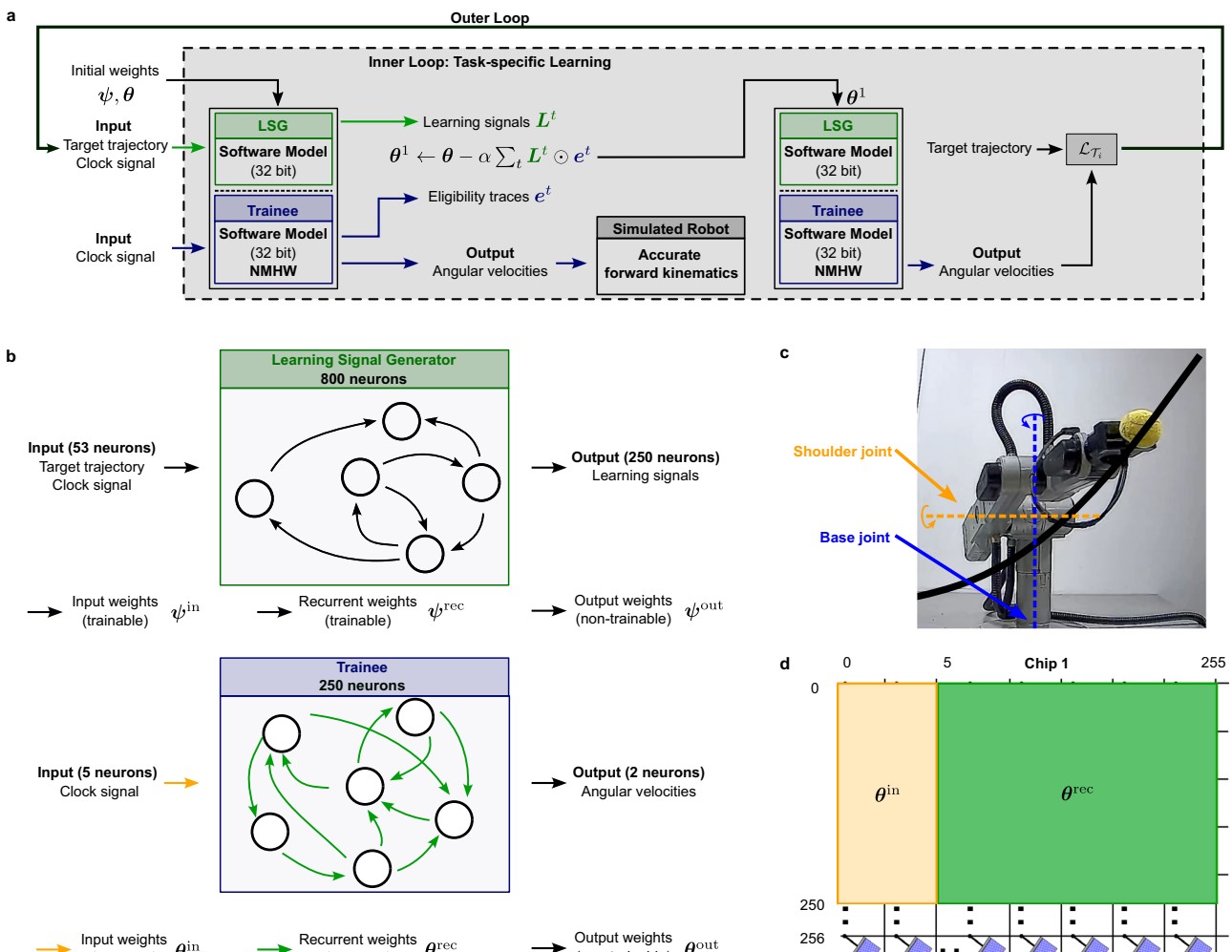

**Fig. 4 | Concept and network architecture for rapid online learning of motor commands with neuromorphic hardware. a** Learning-to-learn setup with natural e-prop. The inner loop consists of two phases. Weight updates are based on learning signals $L^t$ and eligibility traces $e^t$ after the first phase. In the second phase, the produced robot arm trajectory of the updated recurrent SNN is validated against the target trajectory. The resulting error is used for the outer loop update. After meta-training, the software model for the robot is replaced by the real robotic arm (**c**). **b** Network architecture. The network architecture consists of two components, the learning signal generator and the trainee. The trainee produces the motor commands as well as implicitly the eligibility traces. The learning signal generator produces the learning signals which are combined with the eligibility traces to form the gradient updates in the inner loop. **c** A schematic depiction of the robotic arm following a target trajectory indicated with a black line. The joints controlled by the trainee are the base joint marked in blue and the shoulder joint marked in orange. **d** Schematic depiction of the mapping of the input weights $\theta^{\mathrm{in}}$ and the recurrent weights $\theta^{\mathrm{rec}}$ onto the crossbar array structure of the NMHW.

illustrated in blue, the output of the model employing the NMHW in orange and the target in black. One can see that the angular velocities produced by the models are close to zero and the robot arm hardly moves. This indicates that the network does not have any particular prior for a target trajectory. Figure 5b shows the behavior of the model after the one-shot update for one example target trajectory. On the left two panels, one can see the angular velocity output of the networks. To quantify the agreement between the model outputs and the target angular velocities, we evaluated three additional trajectories, see Supplementary Figs. 3, 4, and 5. The root-mean-squared error (RMSE) between the 32 bit model and the target angular velocities for the two joints was $(0.0381 \pm 0.0070)$ rad/s and $(0.0363 \pm 0.0057)$ rad/s respectively. The RMSE of the NMHW model was $(0.1274 \pm 0.0811)$ rad/s and $(0.0668 \pm 0.0079)$ rad/s for the two joints respectively. These small differences indicate a good overlap of the produced angular velocities with their corresponding targets, which can also be observed visually.

The produced trajectory in the Euclidean space is shown on the right. The green curve represents the trajectory produced by the NMHW model which was then executed by the ED-Scorbot. Similar to

the angular velocities, there is a good agreement between the target trajectory and the ones produced by the models as well as by the ED-Scorbot. Supplementary Video 1 shows the robot performing the trajectory shown in Fig. 5b. We measured the RMSE deviation between the target trajectory and the trajectories of the models and computed the mean and standard deviation across the 4 trajectories. The 32 bit model deviated on average by $(2.2136 \pm 1.1003)$ cm, the NMHW model by $(6.6921 \pm 3.3971)$ cm, and the ED-Scorbot by $(7.8242 \pm 2.5005)$ cm. Although the deviation of the NMHW is larger than the one of the software model, its small value is still remarkable given the large reduction of bit precision of weights in a recurrent SNN. For a detailed investigation of the weight distributions of the NMHW model, see Supplementary Fig. 6.

In summary, our results from the second task demonstrate that L2L can be combined with online learning and applied to recurrent spiking neural networks with PCM-based synapses on in-memory computing NMHW to rapidly learn the generation of motor commands. Importantly, a single inner loop update step is sufficient to tune the network for a particular target trajectory, which enables efficient realizations with NMHW.

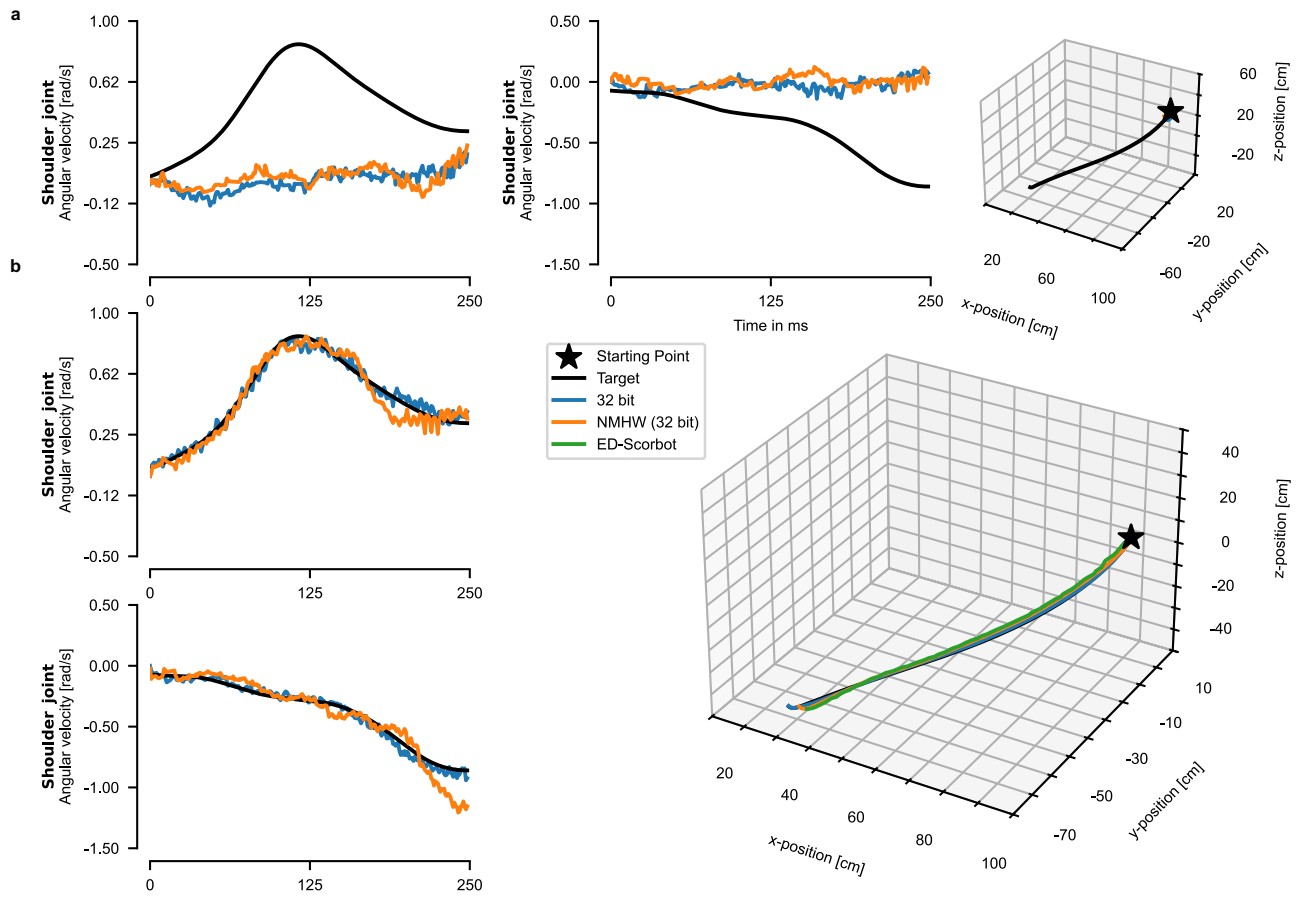

**Fig. 5 | Results of learning motor commands with neuromorphic hardware.**
**a** Angular velocities and trajectories in the Euclidean space of the meta-trained network in software (blue) and with NMHW (orange) before the inner loop update. **b** Angular velocities and trajectories in the Euclidean space of the networks after one-shot learning. The green trajectory shows the trajectory of the ED-Scorbot. The result with the label "NMHW" has been collected employing the NMHW described in Section "Neuromorphic hardware" in "Methods".

## Discussion

In this work, we demonstrated rapid learning on a PCM-based in-memory computing platform through the concept of L2L. As a result, the neuromorphic hardware can not just address a single task, but rather quickly adapt to and solve any instance from a family of related tasks. We showcased its versatility using two different network architectures and tasks.

Firstly, we trained a convolutional neural network with MAML to solve few-shot image classification on the Omniglot dataset. MAML enables the optimization of initial model weights such that a new task can be learned with a small number of online weight updates which makes it particularly appealing for usage on neuromorphic hardware. Moreover, we adapted MAML to further reduce the number of required weight updates by only adjusting the weights of the dense layer. Thus, the weight update on the NMHW reduces to a simple delta learning rule, and only a small number of PCM devices are updated. We have found that the software-trained models ported onto the neuromorphic hardware performed on par with evaluations done solely in software which highlights that accurate hardware models are not necessary in this task. Furthermore, MAML can in principle be applied to any model that is trainable with gradient-based optimization. However, standard methods like BPTT are problematic in the neuromorphic context as they cannot be implemented efficiently on hardware. By training only the weights of the dense layer in the inner loop, we avoided this issue as no backpropagation of errors is necessary in this case. An interesting alternative would be to consider MAML in combination with hardware-friendly learning algorithms such as

e-prop[54], OSTL[37], or OSTTP[15] in the inner loop. Especially variations of OSTL or OSTTP may prove beneficial, as they allow to train multi-layered networks in a fully forward manner, avoiding update locking problems of BPTT that would otherwise hinder the efficient training of the convolutional neural network. With these algorithms, the full potential of the NMHW can be leveraged.

From the biological perspective, meta-training can be interpreted as an evolutionary process that shapes neural circuits of the brain to become efficient learners for behaviorally relevant tasks[19]. It was proposed in ref. 19. That biological learning may rely on three loops: a loop on the time scale of evolutionary processes, a loop on the time scale of the lifetime of the animal, and a fast loop for learning individual tasks. We did not consider the second loop explicitly. In principle, our outer loop could subsume both, the second loop and the first evolutionary loop. Nevertheless, it would be interesting to model a secondary loop explicitly where in particular unsupervised learning could play a prominent role.

In our second task, we explored the biological perspective in greater detail and trained a spiking neural network to produce motor commands to control a robotic arm. In particular, we used natural e-prop to mimic the brain's ability to quickly adapt to new tasks. By co-training a learning signal generator that generates the updates for the synaptic weights of the trainee network, a new target trajectory for robotic arm can be learned with just a single weight update. Similarly to the Omniglot few-shot image classification in our first task, the meta-training was carried out in software without a precise model of the hardware. Before the adaptation phase for evaluation, the trainee

network weights were transferred onto the neuromorphic hardware, and weight updates were performed on the NMHW. The trajectories generated by the network executed on the neuromorphic chip closely matched the trajectories of the high-precision software model both with the simulation and the real robot. Spiking neural networks have complex dynamics compared to feed-forward networks, such as CNNs. Therefore, it is more surprising than in the first task that the inaccuracies of the low-precision analog PCM weights only weakly influenced network performance. It is possible that the learning-to-learn procedure resulted in a robust network that is less prone to variations of the NMHW. However, more analysis would be needed to draw reliable conclusions.

In addition to the initialization-based L2L method investigated in the first task and the parameter-generation-based L2L method of the second task, another interesting direction for future research emerges from the model-based L2L methods. In recent years, memory-augmented neural networks have gained traction in the ML community[25,55] as well as in the SNN community[56]. One of their main advantages compared to other neural networks is the ability to explicitly store information or associations in an external memory and retrieve it at a later time. One can envision that not solely task-specific information is stored in this memory, but rather information that is important for a family of related tasks, which then gets optimized through L2L. However, such memory-based approaches typically require a controller to perform the read and write operations leveraging the external memory, which typically is a (complex) neural network involving complicated update rules, e.g., an LSTM network trained with BPTT. Hence, they may require a powerful CPU to carry out the computations. Nevertheless, in-memory computing architectures based on PCM devices, such as the one that we employed in our work, are well-suited to represent the external memory[57–59], and hence would also present a good candidate for model-based L2L with NMHW.

To conclude, our consistent findings across two tasks demonstrate that L2L can enable PCM-based neuromorphic hardware to rapidly adjust to new tasks with only very few training examples and update steps. This is especially striking in the case of the motor command generation, as there is only a single update involved, which leads to a very light computational load. Notably, our findings underline the robustness of both learning-to-learn frameworks when considering hardware variability, revealing that direct, time-intensive hardware-in-the-loop training, or accurate models of the hardware, can be substituted with simple software approximations without sacrificing performance. Moreover, the capabilities of NMHW and in particular the matrix sizes that they can represent have recently increased significantly[10]. This allows larger and more complex network architectures to be mapped and further boosts the application ranges of L2L on NMHW. Therefore, this work lays the foundation for a promising direction for efficient neural network training on neuromorphic hardware, emphasizing the viability of simulation-based meta-training followed by few on-chip parameter updates.

## Methods

### Neuromorphic hardware

In conventional computing architectures, also referred to as von Neumann architectures, the memory and processing units are separate, necessitating frequent data transfers between them which leads to latency and energy inefficiencies. In contrast, in this work, we employ an analog in-memory neuromorphic hardware. This NMHW is, inspired by the human brain and integrates computation and storage in the same physical location, thus constituting an example of a non-von Neumann computing architecture.

More specifically, the employed NMHW leverages the analog properties of memristive devices, such as PCM and resistive random-access memory, to encode information, such as the weights of a neural network, in their conductance. When these devices are arranged in a crossbar topology, matrix-vector multiplication can be carried out by encoding the matrix elements in the conductance of the devices and the vector elements in voltage stimuli, applied on the rows of the array. According to Ohm's and Kirchhoff's laws, the induced currents on the columns of the array are proportional to the result of the matrix-vector multiplication. This type of computation is highly efficient as it eliminates the need to transfer the matrix elements, it is highly parallelizable, and it takes place in the analog domain. Given the prevalence of matrix multiplications in the majority of contemporary AI workloads, analog in-memory computing emerges as a promising candidate for an efficient AI hardware platform.

In our experiments, we utilize the NMHW platform described in ref. [21]. This platform consists of two PCM-based cores, each featuring a $256 \times 256$ crossbar array. Each unit cell adopts a 4R8T differential configuration, employing two devices to represent positive weights ($BL^+$) and two devices for negative weights ($BL^-$), for a total of 262,144 devices per core. Furthermore, each core incorporates 256 digital-to-analog converters responsible to provide input stimuli to the array employing signed 8 bit pulse-width modulation. In particular, two distinct input lines are used to provide inputs to the devices representing positive ($WL^+$) and negative ($WL^-$) weights, enabling multi-phase MVM operations. In this work, we utilize 4-phase MVM operation, applying only just one sign of inputs to one sign of weights per phase, for increased precision[10]. Finally, 256 analog-to-digital converters are employed to digitize the induced current, alongside a local digital processing unit tasked with performing affine correction post-digitization and converting the output to its 8 bit representation. Each core of the platform operates independently, controlled by an onboard field-programmable gate array (FPGA) module, and is abstracted as an 8 bit IN/8 bit OUT MVM unit for the purposes of this work.

The cores were fabricated at 14 nm, with the PCM inserted in the backend of line at IBM Research at Albany NanoTech. The PCM devices are of mushroom-type and comprise a ring heater for the bottom electrode, doped $Ge_2Sb_2Te_5$ as the phase-change material, and a top electrode film stack, which is subtractively patterned to form the mushroom top. More details can be found in ref. [10]. The conductance of the PCM device is tuned by changing the relative volume of the material in the crystalline and amorphous phases, which correspondingly exhibit high and low conductance. This modulation is achieved through the application of specialized electrical stimuli during the programming phase. Given the highly stochastic nature of this process, an iterative read-write verify algorithm is employed to fine-tune the conductance of the devices. Note that we use two devices to encode a weight, selecting the pair that corresponds to its sign, while the remaining two devices are maintained in a highly resistive RESET state to prevent any current flow. The details of our programming algorithm are elaborated upon in ref. [44]. Additionally, post-programming, the PCM devices are subject to temporal conductance drift. To address this, we implement an affine correction for each column within the local digital processing unit of the core[10].

### Deploying models on the neuromorphic hardware

The deployment of neural networks on our platform is facilitated by an automated software stack, which leverages PyTorch model definitions and its runtime. This stack treats the platform outlined in Section "Neuromorphic hardware" as two distinct 8 bit IN/8 bit OUT MVM units, and allocates all MVM operations to them while performing all other operations on the host machine. The deployment flow followed by the stack is described in the following.

Initially, the model is parsed to identify all layers containing MVM operations. Subsequently, these layers are assigned to the two cores and mapped to distinct regions on their crossbars. For linear layers, the mapping process is straightforward, as we place the weight arrays

without additional processing in the crossbars. In the case of convolutional layers, we adopt the im2col strategy, where the filters are flattened into a single-weight array, and the patches of the input feature maps are transformed per the im2col scheme[60]. In both cases, if the resulting weight array exceeds the size of the crossbar in any dimension, the software stack fragments it into segments that fit within the array (see Fig. 2e). During inference, the stack combines the partial results from the fragmented arrays accordingly.

Following this, the software stack conducts various post-training hardware-related calibration steps to ensure maximum MVM precision[61], and programs all weight arrays onto the two cores. Finally, the stack utilizes the PyTorch runtime to execute each MVM-containing module on the neuromorphic hardware. Modifications to the PyTorch runtime have been implemented to enable parallel execution of layers across the two cores in a pipelined fashion, when permissible by the mapping.

## The ED-Scorbot robotic arm

The ED-Scorbot platform[62] is derived from a modified Scorbot ER-VII commercial robot, and it operates on an event-driven neuromorphic system. These modifications enable the robot to be controlled using spike-based motor controllers. The ED-Scorbot is equipped with six degrees of freedom (DoF), which are generally referred to as joints. Each joint is capable of rotating using a DC motor. This motor is equipped with a dual optical encoder, which is utilized to accurately determine the present location of the joint. The previous control circuitry of the Scorbot ER-VII was replaced by a Zynq-7100 FPGA board[63], optocoupled logic for electromagnetic isolation from the motors, and a new 12 V power supply. This new controller setup on the ED-Scorbot implements six spike-based proportional-integrative-derivative (SPID) controllers[64]. The reference given to the SPID as input can be provided as a digital signal that represents the target position of the respective joint. The main advantage of controlling the robot with spike-based controllers compared to a classic digital controller, is the reduced power consumption and the lower latency[62]. When approaching the target position, the SPID controller will produce less activity the closer the joint is to its commanded position. Ideally, a joint that has reached its desired position will make the SPID controller not fire any spike, until the commanded position is changed.

For the 3D control of the robotic arm, we used the two first joints of the robot (base and shoulder), while all other joints were fixed. Trajectories to be executed by the robot or the robot model were provided by the trainee SNN in the form of angular velocities for the two controlled joints of the robot. These velocities were converted to instantaneous angles for each joint at a time step of the SNN (the time step was 1 ms).

For outer loop training, 3D robot arm trajectories were executed in Python via the forward kinematics model described below (eqs. 8–10). To speed up training, the commanded angular velocities and corresponding instantaneous angles were immediately applied to each joint, i.e., each joint instantaneously reached its commanded position at each time step. This trajectory was then used to calculate the loss function for the outer loop updates.

For evaluation after outer loop training, both the simulated model and the real robot arm were used. For the simulated robot, the angles, calculated from the angular velocities produced by the SNN, were converted to Euclidean coordinates of the end-effector via the forward kinematics model. At each SNN time step, the coordinates were recorded and used for evaluation. The mean-squared error between the target trajectory and the commanded trajectory in Euclidean space was used to evaluate the performance of the SNN.

For evaluation utilizing the physical robot, the angular velocities for the joints were provided to the robot through a file-based protocol. First, they were translated into joint angles and then converted to spike-reference values. Those served as inputs for the SPID which then controlled the joints of the robot. These target joint angles for each SNN time step were applied for 250 ms in order to allow the robot to reach the position. This means that the trajectory which lasted 250 ms in the time scale of the SNN (250-time steps, each 1 ms), lasted approximately one minute in real-time when executed on the physical robot. The measured positions of the robot joints were recorded at every time step and used to compute the Euclidean coordinates of the tip of the arm, employing the forward kinematics of the robot. Again, the mean-squared error between the target trajectory and the commanded trajectory in Euclidean space was used to evaluate the performance of the SNN.

The formulation for the forward kinematics of the ED-Scorbot robotic arm based on the Denavit-Hartenberg (D-H) matrix is given by

$$
\begin{aligned}
x = &\, a_1 c(\theta_1) + a_2 c(\theta_1\theta_2) - a_3 s(\theta_2\theta_3)c(\theta_1) + a_3 c(\theta_1\theta_2\theta_3) \\
&+ a_4(-s(\theta_2\theta_3)c(\theta_1) + c(\theta_1\theta_2\theta_3))c(\theta_4) - d_2 s(\theta_1) \\
&+ a_4(-s(\theta_2)c(\theta_1\theta_3) - s(\theta_3)c(\theta_1\theta_2))s(\theta_4) - d_3 s(\theta_1),
\end{aligned} \tag{8}
$$

$$
\begin{aligned}
y = &\, a_1 s(\theta_1) + a_2 c(\theta_1\theta_2) - a_3 s(\theta_1\theta_2\theta_3) + a_3 s(\theta_1)c(\theta_2\theta_3) \\
&+ a_4(-s(\theta_1\theta_2\theta_3) + s(\theta_1)c(\theta_2\theta_3))c(\theta_4) + d_2 c(\theta_1) \\
&+ a_4(-s(\theta_1\theta_2)c(\theta_3) - s(\theta_1\theta_3)c(\theta_2))c(\theta_4) + d_3 c(\theta_1),
\end{aligned} \tag{9}
$$

$$
\begin{aligned}
z = &\, -a_2 s(\theta_2) - a_3 s(\theta_2)c(\theta_3) - a_3 s(\theta_3)c(\theta_2) + a_4(s(\theta_2\theta_3) \\
&- c(\theta_2\theta_3))s(\theta_4) + a_4(-s(\theta_2)c(\theta_3) - s(\theta_3)c(\theta_2))c(\theta_4) + d_1,
\end{aligned} \tag{10}
$$

where for the sake of clarity and brevity, we represent cos and sin functions with the letters $c$ and $s$, respectively, and multiplications of cosine and sine functions are expressed as in this example: $\cos(\theta_1)\cos(\theta_2) = c(\theta_1\theta_2)$. Table 1 shows D-H parameters for our setup.

## Few-shot image classification

In this work, we demonstrate few-shot image classification with MAML on two datasets. In the first case, we used the Omniglot dataset and followed the architecture outlined in ref. 22. We used a convolutional neural network with four blocks consisting of convolutional layers with $3 \times 3$ convolutions and a stride of 2, followed by a ReLU non-linearity and a batch normalization. The output of these four blocks was passed into a max-pooling layer, with a size of $2 \times 2$ and a stride of 1, followed by a dense output layer with a softmax activation. Compared to[22], the four convolutional layers have been reduced from 64 filters to 56 filters in order to fit the network onto the NMHW described in "Neuromorphic hardware", see Fig. 2d for an illustration of the network configuration. The network was trained using the cross-entropy loss. The meta-training was performed for 30,000 iterations with a batch size of 40 and learning rate $\beta = 0.001$, while the inner loop performed $n = 4$ gradient update steps with learning rate $\alpha = 0.1$. After meta-training, the models were evaluated for 100 tasks. Note that during inner loop training, as well as during the evaluation, only the weights of the dense layer were adapted.

In the second case, we used the more demanding CIFAR100-FS dataset[46], a few-shot learning dataset derived from CIFAR100. For this dataset, we used a convolutional neural network with four blocks consisting of convolutional layers with $3 \times 3$ convolutions, a stride of

**Table 1 | Denavit-Hartenberg parameters of the ED-Scorbot**

| Joint | $\theta_i$ | $d_i$[cm] | $a_i$[cm] | $\alpha_i$ |
|---|---|---|---|---|
| 1 | $\frac{-23.6\pi}{180}$ | 35.85 | 5.0 | $\pi/2$ |
| 2 | $\frac{22\pi}{180}$ | −9.8 | 30 | $\pi$ |
| 3 | $\frac{22.4\pi}{180}$ | 6.5 | 35 | 0 |
| 4 | 0 | 0 | 22 | 0 |

2256 hidden channels, followed by a ReLU non-linearity and a batch normalization. The output of these four blocks was passed into a max-pooling layer with a size of 2 × 2 and a stride of 1. On top of that, we used two dense layers with biases, one with 256 output units and one with 5. Finally, we used a softmax activation on the 5 output units to obtain class probabilities. The network was trained using the cross-entropy loss. The meta-training was performed for 37,000 iterations with a batch size of 4 and learning rate $\beta = 0.001$, while the inner loop performed $n = 5$ gradient update steps with learning rate $\alpha = 0.1$. After meta-training, the models were evaluated for 100 tasks. Note that during inner loop training, as well as during the evaluation, we only updated the weights and biases of the last dense layer. The weights were updated according to eq. (3) and the biases according to

$$\Delta b_l = \alpha \left( y_l^{(d)} - f_{\boldsymbol{\theta}^j, l} \right). \tag{11}$$

Because we realized that the accuracy during inner loop training reached close to 100% already after the fourth gradient update, we also evaluated the same model with only 4 inner loop gradient update steps.

### One-shot learning via natural e-prop

In natural e-prop, a learning signal generator SNN and a trainee SNN operate jointly. While the trainee produces the functional output, the LSG produces learning signals that are employed to form the weight updates of the trainee. During the meta-training phase, the LSG and trainee network are jointly trained on a family of tasks. During this phase, the weights of the LSG initial weights of the trainee are trained using BPTT. In the inner loop, only the weights of the trainee network are adapted utilizing the learning signals emitted by the LSG and the eligibility traces of the trainee.

We considered SNNs for both, the LSG and for the trainee network. The trainee was composed of 250 leaky integrate-and-fire (LIF) neurons, followed by a linear readout layer. It received a clock-like signal that was the same across all trials. The goal of the trainee was to produce motor commands in terms of angular velocities $\boldsymbol{\Phi}^t$ such that the produced trajectory $\hat{y}^t$ by the robotic arm matches the target trajectory $y^t$. See Fig. 4a, b for an overview. The LSG was composed of a mix of LIF and adaptive leaky integrate-and-fire (ALIF) neurons, received the same clock-like signal as the trainee and additionally, the target trajectory $y^t$. It consisted of 800 neurons in total where 30% of the population were ALIF neurons. In contrast to the trainee, the task of the LSG was to produce suitable learning signals, so that after a single update of the weights of the trainee, the target trajectory could be followed.

Both SNNs were simulated in discrete time with a resolution of 1 ms and synaptic delays were fixed to 1 ms. For the LIF neurons, the membrane voltage $v_j^t$ and the presence of output spikes ($z_j^t = 1$) evolved according to

$$v_j^{t+1} = \gamma v_j^t + \sum_{i \neq j} \theta_{ji}^{\text{rec}} z_i^t + \sum_i \theta_{ji}^{\text{in}} x_i^t - z_j^t v_{\text{th}} \tag{12}$$

$$z_j^t = H\left(\frac{v_j^t - v_{\text{th}}}{v_{\text{th}}}\right), \tag{13}$$

where $x_i^t = 1$ indicates an input spike from neuron $i$ at time $t$, $v$th is the spike threshold, $\theta_{ji}^{\text{rec}}$ and $\theta_{ji}^{\text{in}}$ are the weights for recurrent and input neurons between neuron $i$ and neuron $j$, respectively. The membrane decay factor $\gamma$ is defined by $\exp(-\frac{\delta t}{\tau_m})$, where $\delta t$ is the simulation time step and $\tau_m$ is the membrane time constant. The neuronal reset was realized using the term $-z_j^t v_{\text{th}}$. The Heaviside step function $H$ is defined as $H(x) = \mathbf{1}_{x \geq 0}$ and is not differentiable. This can be resolved by

using a pseudo-derivative given by $h_j^t = \lambda \max(0, 1 - |\frac{v_j^t - v_{\text{th}}}{v_{\text{th}}}|)$, where $\lambda$ is a dampening factor that controls the slope of the pseudo-derivative. The eligibility traces of LIF neurons can be written as $e_j^{t+1} = h_j^t \sum_{t' \leq t} \gamma^{t-t'} z_i^{t'}$ which corresponds to a low-pass filtered version of the pre-synaptic spikes. Thus, the weight updates for the recurrent weights and input weights can be formulated as

$$\Delta \theta_{ji}^{\text{rec}} = -\alpha \sum_t L_j^t h_j^t \sum_{t' \leq t} \gamma^{t-t'} z_i^{t'} \tag{14}$$

$$\Delta \theta_{ji}^{\text{in}} = -\alpha \sum_t L_j^t h_j^t \sum_{t' \leq t} \gamma^{t-t'} x_i^{t'}, \tag{15}$$

where the learning signal $L_j^t$ is given by

$$L_j^t = \alpha_e L_j^{t-1} + \sum_i \psi_{ji}^{\text{out}} \xi_i^t. \tag{16}$$

Here, the constant $\alpha_e$ denotes the learning signal decay rate, $\psi_{ji}$ refers to the output weights, and $\xi_i^t$ is the output of neuron $i$ of the learning signal generator. The learning signal can be interpreted as a low-pass filtered version of the learning signal generator output.

For the ALIF neurons, the evolution of the membrane voltage $v_j^t$ is equivalent to the LIF neurons as described above, but the threshold $A_j^t$ is adaptive and evolves according to

$$A_j^t = v_{\text{th}} + \beta a_j^t, \tag{17}$$

where $v$th is the baseline threshold, $\beta$ is the threshold increase constant, and $a_j^t$ is the threshold adaptation given by

$$a_j^{t+1} = \rho a_j^t + H\left(\frac{v_j^t - A_j^t}{v_{\text{th}}}\right), \tag{18}$$

with the decay factor $\rho = \exp(-\frac{\delta t}{\tau_a})$, the discrete-time step $\delta t = 1$ ms, and the adaptation time constant $\tau_a$.

The weight update above describes the inner loop update of the meta-learning procedure. For the outer loop, the initial weights of both the trainee and the learning signal generator were optimized via backpropagation through time. The loss function for the outer loop is given by

$$\mathcal{L}_{\mathcal{T}_i} = \sum_t \left( \frac{1}{2} \left( \hat{y}_{\text{test}}^t - y_{\text{test}}^t \right)^2 + \frac{1}{2} \left( \hat{\boldsymbol{\Phi}}_{\text{test}}^t - \boldsymbol{\Phi}_{\text{test}}^t \right)^2 \right) + \mathcal{L}_{\text{reg}} \tag{19}$$

$$\mathcal{L}_{\text{reg}} = \epsilon \sum_j \left( \left( \frac{\delta t}{T} \sum_t z_j^t \right) - f_{\text{target}} \right)^2, \tag{20}$$

where $\hat{y}_{\text{test}}^t, y_{\text{test}}^t, \hat{\boldsymbol{\Phi}}_{\text{test}}^t$ and $\boldsymbol{\Phi}_{\text{test}}^t$ refer to trajectories and angular velocities at test time after a single weight update of the trainee. The additional loss term $\mathcal{L}_{\text{reg}}$ implements a firing rate regularization[36] with target firing rates of $f_{\text{target}} = 10$ Hz (20 Hz) with regularization coefficients of $\epsilon = 0.25$. $T$ denotes the duration of a single trial.

This optimization problem was solved with the ADAM optimizer, across tasks sampled from a task distribution $\mathcal{F}(\mathcal{T})$. See "Algorithm 2" for a detailed description of the interplay between the meta-training and the inner loop updates. The model was trained for 100,000 iterations with mini-batches of 90 trajectories. The Adam optimizer used a learning rate of 0.0015 with a learning rate decay factor of 0.99 after 500 training iterations and an inner loop learning rate of 0.0001. The membrane time constant of 20 ms, a refractory time of 5 ms, a dampening factor of 0.3, a threshold increase $\beta$ of 1.6, and an adaptation time constant of 600 ms were used for both the LSG and trainee

neurons. A threshold voltage $v$th of 1.3 (0.6) was used for the LSG (trainee) network.

The clock-like input signal was realized using 5 input neurons that fired with 100 Hz for 50 ms each one after another in a sequence for a total of 250 ms. The target angular velocities of the robot arm were generated using a Wiener process $W_t$ where $W_0 = 0$ and $W_{t-u} - W_t \sim \mathcal{N}(0, u)$ at time $t$ with a variance $u$ of 0.09. The realization of the Wiener process was then smoothed using a Hann window $w(n)$

$$w(n) = \frac{1}{2} - \frac{1}{2}\cos\left(\frac{2\pi n}{M-1}\right) \tag{21}$$

with a length of $M = 120$ steps via a convolution $(W * w)[t]$. In addition, safeguards were introduced that prevented the robot arm from performing trajectories that could cause damage to the robotic platform. Based on the forward kinematics model described above in Section "The ED-Scorbot robotic arm" the positions of the end-effector were computed in Euclidean coordinates and passed to the LSG. To convert the Euclidean coordinates into a series of spike trains, the following procedure was carried out: Each dimension of the Euclidean space was discretized into 16 regions. Each region was represented by the activity of a single neuron with a firing rate of 100 Hz. Therefore three neurons were active at any given moment encoding the position of the end-effector.

The use of the NMHW in this task was as follows. After meta-training, the input weights $\boldsymbol{\theta}^{\text{in}}$ and the recurrent weights $\boldsymbol{\theta}^{\text{rec}}$ of the trainee network were mapped onto a single core of the NMHW after meta-training, see Fig. 4d. Non-plastic weights were kept in software. For testing, the hardware was then used to compute all MVMs needed to execute the trainee SNN. After the presentation of the target trajectory in the first 250 ms, weight updates were computed and the corresponding PCM devices on the hardware were updated. Then, the trainee was again executed for 250 ms with the clock input using the NMHW for the MVMs.

## Data availability

The Omniglot data for Fig. 2 and CIFAR100-FS data for Fig. 3 are presented in ref. 33 (https://github.com/brendenlake/omniglot) and in ref. 46 (https://github.com/bertinetto/r2d2?tab=readme-ov-file). The data for the robotic task generated in this study can be re-created by using the Source Code files provided in an open-source repository along with this manuscript[65].

## Code availability

We have published the source code that allows the reader to reproduce our results of the paper under the Apache-2.0 license on GitHub. It can be accessed using[65].

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

## Acknowledgements

This work was funded in part by the CHIST-ERA grant CHIST-ERA-18-ACAI-004 [R.L.], by grant PCI2019-111841-2 funded by MCIN/AEI/ [10.13039/501100011033, A.L-B.], by the Swiss National Science Foundation (SNSF) [20CH21_186999 / 1, A.P. and T.O], by the US National Science Foundation (NSF) EFRI grant [#2318152, R.L.] and by the European Union. Finally, we received support from the "Formación de Profesorado Universitario" Scholarship from the Spanish Ministry of Education, Culture, and Sports, [FPU19/04597, E.P.]. This research was funded in whole, or in part, by the Austrian Science Fund (FWF) [10.55776/I4670-N, R.L.]. For the purpose of open access, the author has applied a CC BY public copyright licence to any Author Accepted Manuscript version arising from this submission. Furthermore, we thank the In-Memory Computing team at IBM for their technical support with the PCM-based NMHW as well as the IBM Research AI Hardware Center. Moreover, we thank Joris Gentinetta for his help with the setup for the robotic arm experiments.

## Author contributions

T.O. designed experiments, performed hardware experiments, contributed to software experiments, analyzed the data, and wrote the paper. H.P. designed experiments, performed software experiments, contributed to hardware experiments, analyzed the data, and wrote the paper. A.V. contributed to hardware experiments. R.R. contributed to software. S.B. contributed to software experiments and robotic setups.

T.L. contributed to software. E.P. contributed to robotic setup and performed robotic experiments. A.L-B. contributed to robotic setup, robotic experiments, and paper writing. A.P. contributed to the design of experiments and paper writing. R.L. conceived the research, contributed to the design of experiments, and wrote the paper.

## Competing interests

The authors declare no competing interests.
