## [Transparent Peer Review file · Nature Communications]

Rapid learning with phase-change memory-based in-memory computing through learning-to-learn

Corresponding Author: Professor Robert Legenstein

Version 0:

Reviewer comments:

Reviewer #1

(Remarks to the Author)

The paper by Ortner et al. presents meta-learning applied to both few-shot image classification and simple robot arm control. The models are deployed on PCM hardware and with an experimental robotic arm. Overall, the work is technically sound, significant, timely, and potentially appealing to a large audience of the journal.

My technical questions are as follows.

1. Regarding MAML, can you justify why PCM is a perfect fit for hardware implementing this training protocol? In the ex situ training, gradient computations are in the digital domain, so PCM is essentially for inference (with efficiency advantages due to in-memory computing). But is there anything beyond that? For in situ training, does PCM have an advantage over a digital computer? I suppose its programming itself is relatively power-hungry and slow compared to alternative emerging memories as well as CMOS.
2. It's mentioned that the meta-training with limited 4-bit precision for outer loop training is unclear. Does this mean the gradient of the outer loop is 4-bit? Why choose 4-bit? Is this based on the PCM multi-level conductance randomness?
3. The meta-training cost is substantial. What's the advantage of MAML for few-shot learning compared to external memory-based approaches?
4. Why is the hardware update limited to the dense layer in the MAML model?
5. Is the PCM used for in-memory computing or near-memory computing?
6. Concerning e-prop-based inner loop optimization, what's the advantage over alternatives like surrogate gradients? Surrogate gradient is currently dominant, especially for large-scale SNN models.

(Remarks on code availability)

No readme found. Suggest to post that to github for convenience.

Reviewer #2

(Remarks to the Author)

This paper applies meta-learning algorithms to PCM-based compute-in-memory (CIM) hardware and achieves simple task incremental learning. Additionally, it combines the meta-learning algorithmic framework with Spiking Neural Network (SNN) to demonstrate real-time control of a robotic arm. The article claims to achieve software-comparable accuracy on the Omniglot task.

Below are some key comments:

The innovation of this paper lies primarily in applying the meta-learning algorithmic framework to PCM-based neuromorphic

hardware, utilizing both CNN and SNN network architectures, and conducting practical demonstrations with a robotic arm. However, while meta-learning itself is widely studied in the ML community, the paper does not optimize the algorithms for the hardware characteristics. Facing PCM hardware, only the quantization process in simple tasks is considered, while other hardware issues such as read noise, IR-drop, and offsets are not considered. Neglecting these issues could potentially lead to decreased accuracy in more complex tasks. Moreover, there exist some related articles applying meta-learning to memristor-based (including PCM, RRAM, etc.) neuromorphic hardware [Ref1,2]. Therefore, I believe this paper is not suitable for submission to NC but may be more appropriate for journals such as Front. Neurosci.

Below are some technical questions:

1. If the paper demonstrates the robotic arm task using actual chips, please present the demonstration system incorporating PCM-based hardware. How the PCM-based hardware controls the robotic arm?
2. The paper chooses the relatively simple task of incremental learning on Omniglot, thus focusing solely on quantization to achieve software-comparable results. Additionally, to reduce the cost of weight rewriting, only the final layer is updated. If demonstrations were performed on datasets like split-CIFAR10, could the proposed algorithm still achieve similar results? I believe methods such as hardware-aware training [Ref3], which consider precise modeling, become crucial in such scenarios.
3. The rapid learning scenarios mentioned in the paper mainly apply to edge computing. However, meta-learning involves both inner and outer loops, requiring numerous iterations. How can rapid learning be ensured? PCM-based hardware here only accelerates the inference phase, with all training computations performed in software. Thus, in real edge computing scenarios, PCM-based hardware needs to be paired with powerful computing units such as CPU to function properly. What proportion of the acceleration and energy efficiency does PCM contribute to the entire meta-learning process? I would suggest the authors conduct a systematic performance evaluation to determine where the advantages of utilizing PCM-based hardware lie in implementing this algorithm compared to traditional GPU and other computing chips.

[Ref1] Bohnstingl, T., Scherr, F., Pehle, C., Meier, K., & Maass, W. (2019). Neuromorphic hardware learns to learn. *Frontiers in neuroscience*, 13, 451952.

[Ref2] Zhang, W., Wang, Y., Ji, X., Wu, Y., & Zhao, R. (2021). ROA: a rapid learning scheme for in-situ memristor networks. *Frontiers in Artificial Intelligence*, 4, 692065.

[Ref3] Rasch, M. J., Mackin, C., Le Gallo, M., Chen, A., Fasoli, A., Odermatt, F., ... & Narayanan, V. (2023). Hardware-aware training for large-scale and diverse deep learning inference workloads using in-memory computing-based accelerators. *Nature communications*, 14(1), 5282.

(Remarks on code availability)

Reviewer #3

(Remarks to the Author)

This paper first reports an application of learning-to-learn to memristor-based in-memory neuromorphic hardware (NMHW). Two application scenarios are demonstrated with the proposed NMHW: a convolutional neural network performing image classification and a spiking neural network generating motor commands for a real robotic arm. This work lays the foundation for a promising direction for efficient neural network training on neuromorphic hardware. The paper is well-written and well-organized, and the results are promising. Some minor comments I would like to provide here for the authors' reference, shown as follows:

- 1) Could you add some discussion about PCM devices in the introduction? Why do you use PCM devices in this work and what is the difference between the PCM device used in this work and the previous work?
- 2) Could you give more details about the PCM chip in the Results section, such as chip design, platform design, fabrication, and measurements?
- 3) I suggest the authors add a comparison table with the previous related works in the Supplementary Information.

(Remarks on code availability)

Version 1:

Reviewer comments:

Reviewer #1

(Remarks to the Author)

I thank the authors for the prompt responses which have well addressed my early concerns. I thus recommend it for publication as is.

(Remarks on code availability)

The repo looks good, with sufficient instructions to run the applications. I didn't run the code but i have gone through some of them.

Reviewer #2

(Remarks to the Author)

Thank you to the authors for providing a detailed response to the previously raised questions. However, there are still some issues that remain unclear:

1. The authors emphasize that only the inner loop updates the last layer, but the outer loop updates the initial weights θ . During the outer loop process, are the weights of the last layer the only ones being updated? If not, would new weights θ obtained after each outer loop iteration require a re-mapping of the weights? If so, would this part generate additional power consumption? Would it impose limitations on speed?
2. In the manuscript, the inner loop is performed on the PCM array, while the outer loop also requires updating the weights θ based on the computed gradients. When considering a complete hardware system based on PCM designed to implement this algorithm, how would the outer loop be realized?
3. Additionally, the manuscript indicates a significant number of iterations, which could result in longer training times. The term "rapid learning" in the title is intended to be comparative—could the authors please clarify what baseline or alternative methods this is being compared against? Is it referring to the MAML algorithm running faster on PCM-based hardware compared to digital chips, or is it suggesting that the MAML algorithm on PCM is faster than other learning algorithms?
4. Regarding the lack of a figure of the PCM chip, other reviewers have expressed their confusion, though the authors have provided an explanation. Does the data labeled NMHW in the paper refer to the results tested on the PCM chip?

(Remarks on code availability)

Reviewer #3

(Remarks to the Author)

The authors have satisfied all my comments and I have no further questions.

(Remarks on code availability)

Version 2:

Reviewer comments:

Reviewer #2

(Remarks to the Author)

The authors have satisfied all my comments. I don't have any more questions

(Remarks on code availability)

This code looks quite complete and includes the experiments in the manuscript. However, I haven't tried to run it.

Point-by-point response to the reviewer's questions

August 2, 2024

Reviewer 1

Reviewer #1 (Remarks to the Author):

The paper by Ortner et al. presents meta-learning applied to both few-shot image classification and simple robot arm control. The models are deployed on PCM hardware and with an experimental robotic arm. Overall, the work is technically sound, significant, timely, and potentially appealing to a large audience of the journal.

Reply: We thank the reviewer for the careful consideration of our manuscript and we appreciate the acknowledgment of the quality of our work. Below we provide point-by-point responses to every question.

My technical questions are as follows.

1. Regarding MAML, can you justify why PCM is a perfect fit for hardware implementing this training protocol? In the ex situ training, gradient computations are in the digital domain, so PCM is essentially for inference (with efficiency advantages due to in-memory computing). But is there anything beyond that? For in situ training, does PCM have an advantage over a digital computer? I suppose its programming itself is relatively power-hungry and slow compared to alternative emerging memories as well as CMOS.

Reply: There are several aspects as to why the PCM devices are a good fit for the learning-to-learn scenarios we present in this work. As the reviewer mentioned correctly, this type of hardware exhibits significant benefits stemming from in-memory computing, where the weights of the network are stationary, and the matrix-vector multiplications can be performed in constant time. The energy-efficiency of the neuromorphic hardware for matrix-vector multiplications is about

9.76 TOPS/W [1]. With further innovations on the crossbar array, at least one order of magnitude better energy-efficiency is projected for CNN-based, LSTM-based and transformer-based neural network architectures, compared to state-of-the-art conventional CMOS-based architectures, such as GPUs [2]. Moreover, it is important to mention that during in situ training, that is the task adaptation phase in our nomenclature, learning takes place as well and the network architecture is updated. However, a core aspect of our approach is that only the last dense layer (1 120 PCM devices) is updated, which is only a tiny fraction of the overall architecture (less than 1% for the Omniglot task). Therefore, the cost of programming during the task adaptation phase is significantly reduced. In addition, the update rules for the network updates are very simple, for example see Eq. 3 of the manuscript, and could potentially be implemented directly on the neuromorphic hardware in future generations. Finally, once the dense layer of the network has been fine-tuned to the new classes, it can perform inference for those classes with high accuracy without reprogramming the PCM devices again. In real applications, this is a very common scenario. Therefore, our learning algorithms present a perfect fit for PCM-based neuromorphic hardware and allows to unleash its advantages. In order to clarify this better, we enhanced the introduction of the revised manuscript, lines 47–58, with a more detailed explanation why our approach is a perfect fit for PCM devices.

2. It's mentioned that the meta-training with limited 4-bit precision for outer loop training is unclear. Does this mean the gradient of the outer loop is 4-bit? Why choose 4-bit? Is this based on the PCM multi-level conductance randomness?

Reply: The prototype neuromorphic hardware used in this work, employs the PCM devices to carry out matrix-vector computations. Due to their analog nature, PCM devices exhibit manufacturing variations, noise and drift effects, which all result in a limited number of conductance levels that can be stored on the PCM devices. The physics of the PCM devices has been studied thoroughly, for example in [3], and it has been shown experimentally, that these devices achieve an equivalent precision of 4 bit [1]. Therefore, we chose 4 bit precision for one of our reference experiments. To be more precise, the weights of the network architecture were quantized to 4 bit, while the rest of the computations was still in 32 bit. However, a central aspect of our work is that we do not rely on hardware-aware training, requiring a very detailed model of all hardware imperfections, but we rather simply use a model trained in with 32 bit floating-point precision and port it to the hardware. Therefore, we only use the experiment with meta-training in 4 bit as a reference. In order to clarify this aspect better, we enhanced the caption of Fig. 2 and explained the reason for using 4 bit weights directly when they are introduced, see lines 338–341 in the revised manuscript.

3. The meta-training cost is substantial. What's the advantage of MAML for few-shot learning compared to external memory-based approaches?

Reply: As the reviewer realized correctly and as we described in our manuscript in Section 2.1, there are several different meta-learning methods. MAML belongs to the class of “initialization-based methods”, while external memory-based approaches belong to the second class of “model-based methods”, such as [4]. In both classes the meta-training phase bears substantial costs, but the meta-training is necessary to ensure high accuracy. Both classes have their advantages and disadvantages, but in conjunction with PCM-based neuromorphic hardware, MAML exhibits unique benefits over external memory-based techniques. Most importantly, during the inference phase MAML requires only few update iterations of the network to be carried out, in our case we use 4 update iterations. In addition, a central aspect of our approach is to only update the last dense layer, in every update iteration, which represents only a tiny fraction of the entire network. Therefore, the number of parameter updates is significantly reduced compared to external memory-based techniques, where it may be difficult to estimate how many cells need to be updated in every iteration. Moreover, external memory-based approaches typically require a controller to perform the read and write operations leveraging the external memory, which typically is a (complex) neural network involving complicated update rules, e.g., an LSTM network trained with BPTT. Hence, they may require a powerful CPU to carry out the computations. On the other hand, because we only update the last layer in our approach, the learning rule reduces to a simple delta learning rule and could potentially be implemented directly on the neuromorphic hardware in future generations. Thus, MAML presents a perfect fit for PCM-based neuromorphic hardware. In order to clarify this aspect better, we enhanced the caption of Fig. 2. Moreover, we expanded our discussion in the revised manuscript with lines 699–705. Finally, we would like to mention that MAML is only one instantiation of a learning-to-learn algorithm that can unleash the benefits of PCM-based neuromorphic hardware. We also showcase that a spiking neural network trained with natural e-prop can leverage PCM devices to enable rapid learning.

4. Why is the hardware update limited to the dense layer in the MAML model?

Reply: The weight update in our setting is limited to the dense layer for multiple reasons. Firstly, we wanted to maintain the benefits of the in-memory neuromorphic hardware as much as possible and thus wanted to avoid extensive updates of all the parameters of the network. With our approach, the number of PCM devices to be updated is less than 1% of the total number of employed PCM devices. Secondly, only updating the last layer alleviates the need to store all activations of the hidden layers and to backpropagate errors through the network. In fact, the learning rule that we employed (see Eq. 3 of the manuscript)

$$\Delta\theta_{lk} = \alpha \left(y_l^{(d)} - f_{\theta_{j,l}} \right) h_k,$$

is very simple and could potentially be implemented directly on the neuromorphic hardware in future generations. Finally, in real world applications the network

may need to classify many more examples of a class before a new class is presented or examples from a previously seen class need to be classified again. Our approach of only training the dense layer would also allow to keep an already trained dense layer on the chip for later use, which would further boost the applicability of the system. In the revised manuscript we clarified why only the dense layer is trained in the MAML model in the caption of Fig. 2, as well as on line 317. Moreover, we also extended the introduction with details why our approach presents a perfect fit for PCM devices on lines 47–58.

5. Is the PCM used for in-memory computing or near-memory computing?

Reply: We employ the PCM-based hardware for in-memory computing. In particular, we map the kernels of the convolutional layer and the weights of the dense layer onto two crossbar arrays, see Fig. 2d and e of the manuscript. Importantly, these matrices remain stationary on the crossbar array. Inputs are provided to the crossbar array and the matrix-vector multiplications are performed in-memory.

6. Concerning e-prop-based inner loop optimization, what’s the advantage over alternatives like surrogate gradients? Surrogate gradient is currently dominant, especially for large-scale SNN models.

Reply: E-prop leverages surrogate gradients, or often also referred to as pseudo-derivatives, in order to resolve the non-differentiability of the Heaviside step function H of SNNs as defined in Eq. 12 and 14 of the manuscript, see also the description of Eq. 1 and 2 of the original paper introducing e-prop [5]. In particular, we used a surrogate gradient of the form $h_j^t = \lambda \max\left(0, 1 - \left|\frac{v_j^t - v_{th}}{v_{th}}\right|\right)$, to calculate the parameter updates as shown in Eq. 14 and 15 of the manuscript. However, typically a full surrogate gradient training approach also requires to backpropagate errors through time, similar to error backpropagation through time (BPTT), which is computationally very expensive and incompatible with neuromorphic hardware implementations. In contrast, e-prop does not require this process and can thus be implemented efficiently on neuromorphic hardware systems [6].

References

- 1 M. Le Gallo, et al., "A 64-core mixed-signal in-memory compute chip based on phase-change memory for deep neural network inference." *Nat. Electron.*, vol. 6, Sept. 2023, pp. 680-93, doi:10.1038/s41928-023-01010-1.
- 2 S. Jain, et al., "A Heterogeneous and Programmable Compute-In-Memory Accelerator Architecture for Analog-AI Using Dense 2-D Mesh," in *IEEE Transactions on Very Large Scale Integration (VLSI) Systems*, vol. 31, no. 1, pp. 114-127, Jan. 2023, doi: 10.1109/TVLSI.2022.3221390.
- 3 S. R. Nandakumar, et al., "Precision of synaptic weights programmed in phase-

- change memory devices for deep learning inference.” IEEE IEDM 2020 pp. 12-18, doi:10.1109/IEDM13553.2020.9371990.
- 4 Santoro, A., Bartunov, S., Botvinick, M., Wierstra, D., & Lillicrap, T. (2016, June). Meta-learning with memory-augmented neural networks. In International conference on machine learning (pp. 1842-1850). PMLR.
 - 5 Bellec, G. et al. A solution to the learning dilemma for recurrent networks of spiking neurons. Nat Commun 11, 3625 (2020).
 - 6 Frenkel, Charlotte and Giacomo Indiveri. "2022 IEEE International Solid-State Circuits Conference (ISSCC)." ReckOn: A 28nm Sub-mm² Task-Agnostic Spiking Recurrent Neural Network Processor Enabling On-Chip Learning over Second-Long Timescales. IEEE, pp. 20-26, doi:10.1109/ISSCC42614.2022.9731734.

Reviewer #1 (Remarks on code availability):

No readme found. Suggest to post that to github for convenience.

Reply: Thank you for notifying us about this. We have added a README to our code repository for convenience and it can be found under <https://github.com/IGITUGraz/RapidLearningInMemoryComputing>

Reviewer 2

Reviewer #2 (Remarks to the Author):

This paper applies meta-learning algorithms to PCM-based compute-in-memory (CIM) hardware and achieves simple task incremental learning. Additionally, it combines the meta-learning algorithmic framework with Spiking Neural Network (SNN) to demonstrate real-time control of a robotic arm. The article claims to achieve software-comparable accuracy on the Omniglot task.

Below are some key comments:

The innovation of this paper lies primarily in applying the meta-learning algorithmic framework to PCM-based neuromorphic hardware, utilizing both CNN and SNN network architectures, and conducting practical demonstrations with a robotic arm. However, while meta-learning itself is widely studied in the ML community, the paper does not optimize the algorithms for the hardware characteristics. Facing PCM hardware, only the quantization process in simple tasks is considered, while other hardware issues such as read noise, IR-drop, and offsets are not considered. Neglecting these issues could potentially lead to decreased accuracy in more complex tasks. Moreover, there exist some related articles applying meta-learning to memristor-based (including PCM, RRAM, etc.) neuromorphic hardware [Ref1,2]. Therefore, I believe this paper is not suitable for submission to NC but may be more appropriate for journals such as Front. Neurosci.

Reply: We thank the reviewer for appreciating the novelty of our work, for providing additional references and for the careful review. While the reviewer summarized the contributions of our work well, we would like to highlight a few central aspects regarding the references mentioned. [1] is only very remotely connected to the present work. Although, it applies learning-to-learn (L2L) concepts to a single-layer SNN represented with neuromorphic hardware, it demonstrates only very basic Markov decision processes and a bandit task. Furthermore, it does not use any of the more advanced L2L methods, such as MAML or natural e-prop, or more complicated tasks, such as Omniglot or the robotic task, that we leverage in this work. As a side note, the first author of this work, Thomas Ortner (former Bohnstingl), was also the first author of [1]. In contrast, [2] is more closely related to our present work in the sense that it applies MAML and also demonstrate its capabilities on the Omniglot dataset. However, [2] relies on a software simulator for memristor networks that mimics non-idealities of the PCM devices, such as the write noise, to perform the experiments. We couldn't find experimental data where the authors employ physical neuromorphic hardware to run their network, e.g., crossbar array with PCM devices. Additional steps are also used in [2] to improve the network performance, such as regularization constraints or learning rate adjustments. As the authors show in Fig. 5, all these components seem to play a critical role for their network. However, the hardware-specific aspects may be challenging to obtain in real

applications, as they highly depend on the individual devices, e.g. device statistics need to be collected from a large number of devices. It is a central novelty of our work to not rely on hardware-aware training and instead leverage the power of L2L to adjust the network accordingly. In particular, our results demonstrate that the meta-training phase can take place in software with 32 bit precision and thus a network architecture trained in a standard way could be used. Furthermore, another key novelty is that we demonstrate L2L on a physical PCM-based in-memory computing neuromorphic hardware and we show a good performance match between software and hardware. Moreover, in [2] the entire network appears to get updated during the meta-training and the task-adaptation phase. In contrast to this, in our work we only train a minimal fraction of the network (the last dense layer), as weight updates of PCM devices incur energy and latency costs. Thus, we meta-train the entire network architecture in software, but only train the last dense layer during the task-adaptation phase, and we demonstrate results in pure software, as well as with physical neuromorphic hardware. Finally, [2] does not demonstrate their L2L approach with an SNN, while we dedicate the second half of our manuscript to demonstrate that our technique can also be applied to SNNs. Irrespective of those critical differences, we thank the reviewer for bringing this work to our attention and of course we cite and discuss it on lines 71–79 in our revised manuscript. In the following, we also provide point-by-point responses to the individual technical questions.

Below are some technical questions:

1. If the paper demonstrates the robotic arm task using actual chips, please present the demonstration system incorporating PCM-based hardware. How the PCM-based hardware controls the robotic arm?

Reply: In our work we used a PCM-based neuromorphic hardware and a real ED-Scorbot robot. The ED-Scorbot employs a spike-based PID controller, and we provide the details of the robot and its controller in Section 4.3 of the manuscript. On a high-level, the procedure of the robotic task can be seen in Fig. 4a of the manuscript. In particular, this task is based on a feedforward control problem (open-loop), in which we first computed the initial angular velocities for the shoulder and base joints for the ED-Scorbot using the neuromorphic hardware. Afterwards, we utilize a file-based protocol to transfer the angular velocities from the neuromorphic hardware to the ED-Scorbot. The angular velocities are then converted into instantaneous angles for each joint. The spiking PID controller is then tasked to follow the directed angles as accurately as possible. Once the robot finished the initial trajectory, we update the last dense layer of the neuromorphic hardware, produce another set of angular velocities, the robot converts it to instantaneous angles and finishes the final trajectory. We clarified this procedure in the revised manuscript on lines 889–894. The target angular velocities from the hardware, the measured angular velocities from the robot, as well as the executed trajectory from the robot are illustrated in Fig. 4e and f of the manuscript.

2. The paper chooses the relatively simple task of incremental learning on Omniglot, thus focusing solely on quantization to achieve software-comparable results. Additionally, to reduce the cost of weight rewriting, only the final layer is updated. If demonstrations were performed on datasets like split-CIFAR10, could the proposed algorithm still achieve similar results? I believe methods such as hardware-aware training [Ref3], which consider precise modeling, become crucial in such scenarios.

Reply: The Omniglot dataset is a very widely used benchmark dataset to test few-shot learning capabilities [4,5,6]. Therefore, we use it in the first half of our manuscript to demonstrate few-shot learning. However, it is important to reiterate that, in contrast to several related works on Omniglot, we only update the last dense layer during the task-adaptation phase on purpose, because of the many benefits it has for the neuromorphic hardware. It is also essential to emphasize that no sophisticated hardware-aware training schemes were necessary, like the ones mentioned in [2,3]. In fact, it is also a key feature of our approach that no hardware-aware training schemes are required. We used standard 32 bit software pre-training for our hardware results and included the 4 bit quantized pre-training only for comparison. Strikingly with our seemingly simple setup, we were able to demonstrate high accuracy in this task. Furthermore, we found that if we use 4 bit weight quantization during meta-training, the results are on-par to the 32 bit training.

Nevertheless, we acknowledge that there may be more complex datasets than Omniglot to demonstrate few-shot learning and we were eager to investigate our approach in these scenarios as well. Unfortunately, the suggested split-CIFAR dataset [7] has been proposed for incremental, or continual learning, which is different from few-shot learning that we are demonstrating. Instead, we have performed additional experiments on the CIFAR100-FS dataset [8], a few-shot learning dataset derived from CIFAR100 [9]. For this more complex dataset, a bigger network architecture is required [8]. In particular, we use a CNN with 4 convolutional layers with 256 hidden channels, followed by two dense layers, one with 256 output units and one with 5. We adopted the very same learning algorithm as for the Omniglot dataset, and again only updated the final dense layer, with 5 output units, in the inner loop. We used 37 000 iterations in the outer loop, a meta batch-size of 4 and 5 gradient steps. We achieved a mean test accuracy of 71.0% over 100 iterations in software for 5-way 5-shot classification, which is close to the test accuracy of 71.5% reported in [8], where a larger network architecture was used, and all layers were updated during the inner loop. The employed network architecture was too big to fit onto our prototype neuromorphic hardware and hence we resorted to a hardware-accurate emulator to carry out the experiments. This emulator was calibrated on one million PCM devices [10,11] and models all their major non-idealities, such as various noise effects. Therefore, it closely matches the behavior of the neuromorphic hardware that we utilized, and several works have already demonstrated equivalent accuracy between software, the emulator and the real hardware [10,11]. When porting the trained network with standard

32 bit onto the emulated hardware, the network achieved a mean test accuracy of 70.0%, which is very close to the software baseline. These additional results indicate that our approach is also applicable for more complex datasets, even without hardware-aware training and when only the last layer is trained. We have included the additional results into the revised manuscript as a new Fig. 3, along with the description on lines 395–446 and the dataset details on lines 937–960.

3. The rapid learning scenarios mentioned in the paper mainly apply to edge computing. However, meta-learning involves both inner and outer loops, requiring numerous iterations. How can rapid learning be ensured? PCM-based hardware here only accelerates the inference phase, with all training computations performed in software. Thus, in real edge computing scenarios, PCM-based hardware needs to be paired with powerful computing units such as CPU to function properly. What proportion of the acceleration and energy efficiency does PCM contribute to the entire meta-learning process? I would suggest the authors conduct a systematic performance evaluation to determine where the advantages of utilizing PCM-based hardware lie in implementing this algorithm compared to traditional GPU and other computing chips.

Reply: We assume that with the term “inference” the reviewer is referring to the task-specific adaptation phase and we would like to clarify that in this phase the last dense layer of the network architecture is updated and hence it includes learning as well. It is true that in the meta-training phase, training is performed in software, and the PCM hardware does not accelerate training there. However, in an application at the edge, the learning is performed after meta-training in the task-specific adaptation phase. Here, the specific design of our approach is critical because only the last dense layer gets updated and the resulting update rule is very simple, as can be seen in Eq. 3 of the manuscript. Therefore, no powerful computing unit is necessary and in fact, the emerging delta rule could potentially be implemented directly on the neuromorphic hardware in future generations. Moreover, only a tiny fraction of the entire network gets updated, less than 1%, and thus most of the PCM devices remain unchanged during task-adaptation. This allows to leverage the full benefits of acceleration and energy-efficiency advantages stemming from PCM devices. Moreover, in real applications it is likely that multiple examples from the same classes need to be classified before new classes appear, in which case not even an update of a single PCM device is needed. Thus, our approach does not compromise the advantages PCM-based neuromorphic hardware and presents a perfect fit for it. In the revised manuscript we included more details why our approach is perfectly suited for PCM-based neuromorphic hardware on lines 47–58.

The energy-efficiency of the neuromorphic hardware has been studied extensively. In particular, the matrix-vector multiplications offer an energy-efficiency of about 9.76 TOPS/W [12]. With further innovations on the crossbar array, at least one order of magnitude better energy-efficiency is projected for CNN-based, LSTM-based

and transformer-based neural network architectures, compared to state-of-the-art conventional CMOS-based architectures, such as GPUs [13]. Paired with the advantages of our training approach above, L2L with neuromorphic hardware forms a very energy-efficient and capable system.

References

- 1 Bohnstingl, T., Scherr, F., Pehle, C., Meier, K., & Maass, W. (2019). Neuromorphic hardware learns to learn. *Frontiers in neuroscience*, 13, 451952.
- 2 Zhang, W., Wang, Y., Ji, X., Wu, Y., & Zhao, R. (2021). ROA: a rapid learning scheme for in-situ memristor networks. *Frontiers in Artificial Intelligence*, 4, 692065.
- 3 Rasch, M. J., Mackin, C., Le Gallo, M., Chen, A., Fasoli, A., Odermatt, F., ... & Narayanan, V. (2023). Hardware-aware training for large-scale and diverse deep learning inference workloads using in-memory computing-based accelerators. *Nature communications*, 14(1), 5282.
- 4 Lake, Brenden M. et al. "One shot learning of simple visual concepts." *Cognitive Science* 33 (2011): n. pag.
- 5 Vinyals, Oriol, et al. 'Matching Networks for One Shot Learning'. *Advances in Neural Information Processing Systems*, edited by D. Lee et al., vol. 29, Curran Associates, Inc., 2016, file/90e1357833654983612fb05e3ec9148c-Paper.pdf.
- 6 Snell, Jake, et al. 'Prototypical Networks for Few-Shot Learning'. *Advances in Neural Information Processing Systems*, edited by I. Guyon et al., vol. 30, Curran Associates, Inc., 2017,
- 7 Zenke F, Poole B, Ganguli S. Continual Learning Through Synaptic Intelligence. *Proc Mach Learn Res*. 2017;70:3987-3995. PMID: 31909397; PMCID: PMC6944509.
- 8 Bertinetto, L., Henriques, J. F., Torr, P., & Vedaldi, A. (2018). Meta-learning with differentiable closed-form solvers. In *International Conference on Learning Representations*.
- 9 Krizhevsky, Alex. "Learning Multiple Layers of Features from Tiny Images." (2009).
- 10 Joshi, Vinay, et al. "Accurate deep neural network inference using computational phase-change memory." *Nat. Commun.*, vol. 11, no. 2473, 18 May. 2020, pp. 1-13, doi:10.1038/s41467-020-16108-9.
- 11 Le Gallo, Manuel, et al. "Using the IBM analog in-memory hardware acceleration kit for neural network training and inference." *APL Mach. Learn.*, vol. 1, no. 4,

- 1 Dec. 2023, doi:10.1063/5.0168089.
- 12 M. Le Gallo, et al. "A 64-core mixed-signal in-memory compute chip based on phase-change memory for deep neural network inference." *Nat. Electron.*, vol. 6, Sept. 2023, pp. 680-93, doi:10.1038/s41928-023-01010-1.
- 13 S. Jain et al., "A Heterogeneous and Programmable Compute-In-Memory Accelerator Architecture for Analog-AI Using Dense 2-D Mesh," in *IEEE Transactions on Very Large Scale Integration (VLSI) Systems*, vol. 31, no. 1, pp. 114-127, Jan. 2023, doi: 10.1109/TVLSI.2022.3221390.

Reviewer #2 (Remarks on code availability):

Reviewer 3

Reviewer #3 (Remarks to the Author):

This paper first reports an application of learning-to-learn to memristor-based in-memory neuromorphic hardware (NMHW). Two application scenarios are demonstrated with the proposed NMHW: a convolutional neural network performing image classification and a spiking neural network generating motor commands for a real robotic arm. This work lays the foundation for a promising direction for efficient neural network training on neuromorphic hardware. The paper is well-written and well-organized, and the results are promising.

Reply: We thank the reviewer for the acknowledgment of our research contributions and for the carefully crafted review. Below, we provide point-by-point responses to the reviewers' comments.

Some minor comments I would like to provide here for the authors' reference, shown as follows:

1. Could you add some discussion about PCM devices in the introduction? Why do you use PCM devices in this work and what is the difference between the PCM device used in this work and the previous work?

Reply: In this work we demonstrate rapid learning by pairing learning-to-learn techniques with a PCM-based neuromorphic hardware. The PCM devices provide several advantages, such as a high energy-efficiency or accelerated execution of vector-matrix multiplications. Our learning approach presents a perfect fit for PCM devices for several reasons. Firstly, during the task-specific adaptation phase only a tiny fraction of the network architecture is updated. For example, in case of the Omniglot task, only less than 1% of the network architecture is changed. Secondly, the learning rule reduces to a simple delta rule that could potentially be implemented directly on the neuromorphic hardware in future generations. Finally, because of the non-volatile nature of the PCM devices, the system can perform inference for already observed classes for extended periods of time, without ever altering the state of the PCM devices. Therefore, our learning approach suits PCM devices perfectly and allows to harness their potential. In order to clarify this better, we enhanced the introduction of the revised manuscript, lines 47–58, with a more detailed explanation why our approach is a perfect fit for PCM devices.

Furthermore, to the best of our knowledge there is no related work that demonstrates learning-to-learn with physical PCM devices. The prototype hardware leveraged in this work was first presented in [1]. More details about the PCM devices can be found in the methods and the extended data of [2], a work that

discloses a larger chip that also uses the same devices and fabrication process. We also extended the description in Section 4.2 with more details about the PCM devices, specifically on lines 781–787 of the revised manuscript.

2. Could you give more details about the PCM chip in the Results section, such as chip design, platform design, fabrication, and measurements?

Reply: The prototype hardware leveraged in this work was first presented in [1]. Since in this article we are not focusing on the hardware, but on learning paradigms that enable the efficient use of the hardware, we decided not to include details about the PCM chip in the Results section. More details about the PCM devices can be found in the Methods and the extended data of [2], a work that discloses a larger chip that also uses the same devices and fabrication process. We also extended the description in Section 4.2 with more details about the PCM devices, specifically on lines 781–787 of the revised manuscript.

3. I suggest the authors add a comparison table with the previous related works in the Supplementary Information.

Reply: To the best of our knowledge there is no related work that demonstrates learning-to-learn with physical PCM devices. Nevertheless, we agree that a tabular comparison to previous approaches is useful. We therefore added Table 1 to the Supplementary Information that compares our work with the closest related works on learning-to-learn with neuromorphic hardware.

Reviewer #3 (Remarks on code availability):

References

- 1 Khaddam-Aljameh, R., et al. "2021 Symposium on VLSI Technology." HERMES Core – A 14nm CMOS and PCM-based In-Memory Compute Core using an array of 300ps/LSB Linearized CCO-based ADCs and local digital processing. IEEE, pp. 13-19, ieeexplore.ieee.org/document/9508706.
- 2 M. Le Gallo, et al. "A 64-core mixed-signal in-memory compute chip based on phase-change memory for deep neural network inference." *Nat. Electron.*, vol. 6, Sept. 2023, pp. 680-93, doi:10.1038/s41928-023-01010-1.

Point-by-point response to the reviewer's questions

December 4, 2024

Reviewer 1

Reviewer #1 (Remarks to the Author):

I thank the authors for the prompt responses which have well addressed my early concerns. I thus recommend it for publication as is.

Reply: We would like to express our gratitude to the reviewer for the effort of a second review and also thank the reviewer again for the initial suggestions, which we believe enhanced our manuscript.

Reviewer #1 (Remarks on code availability):

The repo looks good, with sufficient instructions to run the applications. I didn't run the code but i have gone through some of them.

Reviewer 2

Reviewer #2 (Remarks to the Author):

Thank you to the authors for providing a detailed response to the previously raised questions. However, there are still some issues that remain unclear:

1. The authors emphasize that only the inner loop updates the last layer, but the outer loop updates the initial weights Θ . During the outer loop process, are the weights of the last layer the only ones being updated? If not, would new weights Θ obtained after each outer loop iteration require a re-mapping of the weights? If so, would this part generate additional power consumption? Would it impose limitations on speed?

Reply: We thank the reviewer for the review and the valuable questions. We will address this and the subsequent question jointly below.

2. In the manuscript, the inner loop is performed on the PCM array, while the outer loop also requires updating the weights Θ based on the computed gradients. When considering a complete hardware system based on PCM designed to implement this algorithm, how would the outer loop be realized?

Reply: We would like to revisit the overall training strategy that we propose in our manuscript, which we acknowledge, may have not been laid out sufficiently clear. In order to improve, we have extended Figure 1 with a new panel **a**, that illustrates our training strategy in greater detail. For your convenience, we have also included the updated Figure 1 below. Our strategy includes two phases, the “meta-training phase” and the “adaptation phase”, see Figure 1a. In the meta-training phase, we perform meta-training of the model fully in software, see left part of Figure 1a. Specifically, all weights of the neural network are kept in software, indicated with green color, and updated, indicated with dashed rectangle. Importantly, the meta-training can be considered as a preparatory step, that is only executed once in software and thus the number of iterations of the outer loop does not impose speed or power limitations. Furthermore, during this phase, no operation is executed on the NMHW.

After meta-training, the network is deployed to the NMHW, where all weights of the network are mapped to the hardware, see yellow color in the right part of Figure 1a. In fact, the same meta-trained network can be deployed to multiple hardware instances, each potentially tackling a different task. Thus, the energy-efficiency and learning speed is of high importance. In the adaptation phase, the task-specific learning is performed on the NMHW, which only involves the execution of the inner loop. In particular, there is no execution of the outer loop, and only a small fraction of the networks’ weights are updated, see dashed

rectangle in the right part of Figure 1a. For example, only the weights of the last dense layer in case of the Omniglot task are updated, while the majority of the weights remain unchanged. Therefore, in every inner loop iteration only 1 120 PCM devices of a total of 342 720 PCM devices are updated. However, it is important to note that although only a tiny fraction of the weights is updated, the NMHW employs in-memory matrix-vector multiplications utilizing all network weights. This accelerates the computations of the network output, and improves the energy-efficiency of the model.

Since there is no outer loop execution, no re-mapping of the entire neural network is required. Additionally, only a few iterations of the inner loop are already sufficient to achieve high performance for a specific task, see for example Figure 2h and Figure 3e of the main manuscript. This combination ultimately enables the rapid and efficient task-specific learning, which is the central goal of our training strategy. We have replaced Figure 1 of the main manuscript with the extended version described above, and include more explanatory text to the legend as well as on lines 154-175.

3. Additionally, the manuscript indicates a significant number of iterations, which could result in longer training times. The term “rapid learning” in the title is intended to be comparative – could the authors please clarify what baseline or alternative methods this is being compared against? Is it referring to the MAML algorithm running faster on PCM-based hardware compared to digital chips, or is it suggesting that the MAML algorithm on PCM is faster than other learning algorithms?

Reply: As already outlined in our response above, our training strategy encompasses two phases, the meta-training phase and the adaptation phase. The significant number of iterations (Figure 2f and Figure 3c), referred by the Reviewer, correspond to the meta-training phase. Since this phase is executed only once as a preparatory step and is done in pure software (no PCM hardware involved), there are no constraints on the speed of execution. The term “rapid learning” refers to the adaptation phase, in which our employed learning algorithms can learn a new task with just very few update steps. For example, MAML for the Omniglot and the CIFAR100-FS task only requires about 4-5 update steps, see Figure 2h and Figure 3e. Therefore, with “rapid learning”, we mean that meta-trained network can perform task-specific adaptation with only a few gradient steps.

Moreover, this is a property of our employed learning algorithms, and as shown in Figure 2h and Figure 3e, it is also applicable for realizations with the NMHW. To further elaborate on the rapid learning property, we conducted an experiment with backpropagation (BP) as an alternative algorithm in the adaptation phase. Even though BP does not employ a meta-training phase, we wanted to compare its performance for task-specific adaptation. We use the identical setup to classify Omniglot images, as presented in Figure 2d of the main manuscript and evaluate our meta-trained network, as well as a network trained from scratch with BP. In

particular, we first train the networks on the 25 train images and afterwards test them on the test images.

The result is shown in Figure 2 below. Even though all models learn to classify the 25 train images very quickly, i.e., with only very few gradient steps (Figure 2a), the BP-trained network exhibits significant overfitting leading to poor generalization performance to the test images (Figure 2b). Note that we explored various learning rates for BP, but the overall result remained the same in all cases. In addition, during our task adaptation phase only a tiny fraction of all the weights of the neural network is updated (280 weights $\hat{=}$ 1 120 PCM devices), whilst with BP all weights of the neural network are updated (85 680 weights $\hat{=}$ 342 720 PCM devices). The combination of updating only a fraction of the neural networks' weights and the better generalization to test images achieved in very few gradient steps, demonstrates that our approach enables rapid learning of new tasks on NMHW. We have added the description of this finding to the main manuscript in lines 405-414 and added to Figure to the Supplementary Information as Supplementary Fig. 2.

4. Regarding the lack of a figure of the PCM chip, other reviewers have expressed their confusion, though the authors have provided an explanation. Does the data labeled NMHW in the paper refer to the results tested on the PCM chip?

Reply: Yes, all data labeled with NMHW in the main manuscript has been collected using the physical hardware platform with the PCM chip. In particular, for the experiments labeled NMHW in Fig. 2, Fig. 3 and Fig. 4, we utilized the neuromorphic hardware platform published in [1, 2]. In the previous round of reviews, we have extended the description of our hardware experiments, see Section 4.1 of the main manuscript. To further clarify this aspect, we have enhanced the captions of Figures 2, 3 and 4 of the main text with a note that the results labeled with "NMHW" have been collected with the neuromorphic hardware described in Section 4.1

References

- 1 R. Khaddam-Aljameh et al., "HERMES Core – A 14nm CMOS and PCM-based In-Memory Compute Core using an array of 300ps/LSB Linearized CCO-based ADCs and local digital processing," 2021 Symposium on VLSI Technology, Kyoto, Japan, 2021, pp. 1-2.
- 2 Le Gallo, M., Khaddam-Aljameh, R., Stanisavljevic, M. et al. A 64-core mixed-signal in-memory compute chip based on phase-change memory for deep neural network inference. *Nat Electron* 6, 680–693 (2023). <https://doi.org/10.1038/s41928-023-01010-1>

Reviewer #2 (Remarks on code availability):

Figure 1: (Caption on next page.)

Figure 1: **Overview of learning-to-learn with neuromorphic hardware.** **a** Schematic depiction of the general training strategy. Left: The meta-training phase is performed with all network weights kept in software, indicated with green color, and involves many iterations. During meta-training, all weights of the network architecture are updated, as indicated by the dashed rectangle. Right: After meta-training, the model is deployed onto, potentially many, neuromorphic hardware instances, where the weights are located on the hardware as indicated by the yellow color. Then, the adaptation phase is carried out, performing task-specific adaptation involving only a few iterations of a simple learning procedure. Importantly, this process only updates a fraction of the weights of the network architecture, indicated by the dashed rectangle, and the updates are performed on the neuromorphic hardware, without requiring a re-mapping of the entire network. **b** The general structure of meta-learning approaches used in this article. The inner loop learning is indicated by the gray box. The input to the inner loop is an initial parameter setting θ and task inputs from a task \mathcal{T}_i . Based on these data points, a neural network model is updated n times. In our settings, these updates were performed on a subset of the model parameters θ . The outer loop chooses in every iteration a new task \mathcal{T}_i from the task family $\mathcal{F}(\mathcal{T})$, runs the inner loop, and updates the initial parameters θ based on the errors in the inner loop. The goal is to find initial parameters θ such that a few inner loop updates lead to good results on any task from $\mathcal{F}(\mathcal{T})$. **c** Unrolled meta-learning procedure that highlights the differences between task-specific adaptation of weights in the inner loop and the meta-parameters in the outer loop. **d** Schematic depiction of a phase-change memory device and its inner working. Information is stored in the phase configuration of the material and electrical pulses can be used to switch between the amorphous and the crystalline phase. **e** The employed neuromorphic hardware comprises a crossbar array structure where at each intersection four PCM devices (4R) and eight control transistors (8T) are located. Two PCM devices represent positive weights, bitline positive (BL⁺), and two represent the negative weights, bitline negative (BL⁻). The weights of a neural network are then mapped onto the crossbar structure and network inputs are provided to the positive devices using (WL⁺) and to the negative devices using (WL⁻).

Figure 2: **Comparison of task-adaptation with a MAML-meta-trained network and a network that is trained from scratch with BP.** **a** Train accuracy during task-adaptation on the 5-way 5-shot Omniglot classification experiment as outlined in Section 2.2 of the main manuscript. The plot shows the evolution of the train accuracy for our model on the NMHW with 32 bit (green curve), on the NMHW with 4 bit (purple curve) and for the from-scratch trained network with BP (red curve). The triangular markers on top of the panel indicate the gradient step at which the train accuracy exceeds 97%. **b** Test accuracy for 25 unseen test images from the Omniglot dataset of the models from **a** after training. While our models on the NMHW are able to classify the test images almost perfectly, test accuracy of around 98%, the network trained with BP struggles to generalize and only achieves a test accuracy of around 20%.

Reviewer 3

Reviewer #3 (Remarks to the Author):

The authors have satisfied all my comments and I have no further questions.

Reply: We would like to express our gratitude to the reviewer for the effort of a second review and also thank the reviewer again for the initial suggestions, which we believe enhanced our manuscript.